# Tafazzin regulates neutrophil maturation and inflammatory response

Przemysław Zakrzewski[1,5], Christopher M Rice[1,5], Kathryn Fleming[1], Drinalda Cela [1], Sarah J Groves[1], Fernando M Ponce-Garcia[1], Willem Gibbs[1], Kiran Roberts[2], Tobias Pike[1], Douglas Strathdee[3], Eve Anderson[3], Angela H Nobbs [4], Ashley M Toye [2], Colin Steward[1] & Borko Amulic [1✉]

## Abstract

**Barth syndrome (BTHS) is a rare genetic disease caused by mutations in the *TAFAZZIN* gene. It is characterized by neutropenia, cardiomyopathy and skeletal myopathy. Neutropenia in BTHS is associated with life-threatening infections, yet there is little understanding of the molecular and physiological causes of this phenomenon. We combined bone marrow analysis, CRISPR/Cas9 genome editing in hematopoietic stem cells and functional characterization of circulating BTHS patient neutrophils to investigate the role of *TAFAZZIN* in neutrophils and their progenitors. We demonstrate a partial cell intrinsic differentiation defect, along with a dysregulated neutrophil inflammatory response in BTHS, including elevated degranulation and formation of neutrophil extracellular traps (NETs) in response to calcium flux. Developmental and functional alterations in BTHS neutrophils are underpinned by perturbations in the unfolded protein response (UPR) signaling pathway, suggesting potential therapeutic avenues for targeting BTHS neutropenia.**

**Keywords** Mitochondria; Neutrophil; Neutropenia; Inflammation; Tafazzin
**Subject Categories** Development; Immunology; Stem Cells & Regenerative Medicine

## Introduction

Barth syndrome (BTHS) is a rare X-linked genetic disease characterized principally by dilated cardiomyopathy, skeletal myopathy, and neutropenia (Clarke et al, 2013; Saric et al, 2015; Taylor et al, 2022). Mutations in the *TAFAZZIN* gene, encoding a mitochondrial lipid transacylase, are the primary cause of BTHS (Bione et al, 1996). Tafazzin is essential for remodeling and maturation of cardiolipin (CL), a major phospholipid of the mitochondrial inner membrane, which has a crucial role in maintaining mitochondrial structure and function (Paradies et al, 2019; Saric et al, 2015). *TAFAZZIN* mutations

lead to an accumulation of an intermediate CL species, monolysocardiolipin (MLCL), impairing mitochondrial metabolism and contributing to defects in cardiomyocytes and skeletal muscle, which have abundant mitochondria and are dependent on efficient oxidative respiration (Greenwell et al, 2022). Apart from regulating ATP generation, CL has been implicated in signaling (Patil and Greenberg, 2013; Pizzuto and Pelegrin, 2020), control of apoptosis (Dudek, 2017; Pizzuto and Pelegrin, 2020), generation of mitochondrial reactive oxygen species (ROS) (Dudek, 2017), and regulation of calcium homeostasis (Ghosh et al, 2020). Furthermore, mitochondria are important regulators of inflammation, and it has been suggested that dysregulated inflammatory response is a component of BTHS (Wilson et al, 2012), although there is a major lack of knowledge on the innate immune response in patients. Neutropenia and neutrophil-mediated inflammation are possible targets for therapeutic interventions, to improve patient wellbeing and survival.

Neutropenia is detected in ~84% of BTHS patients and shows various patterns (Clarke et al, 2013; Steward et al, 2019; Taylor et al, 2022), including intermittent and unpredictable, chronic or severe, or truly cyclical (Steward et al, 2019). Neutrophils kill microbes by phagocytosis, generation of ROS, release of antimicrobial proteins stored in secretory granules, and formation of neutrophil extracellular traps (NETs), which consist of externalized chromatin decorated with antimicrobial proteins (Mayadas et al, 2014). NETs can trap microbes and prevent their dissemination but are also sensed by other innate and adaptive immune cells, leading to proinflammatory cytokine production and propagation of inflammation (Amulic et al, 2012). Importantly, neutropenia in BTHS is associated with various bacterial infections, which are the second leading cause of death in these patients (Clarke et al, 2013).

Despite the prominence of neutropenia in BTHS patients, there is little understanding of the molecular and physiological causes of this phenomenon (Steward et al, 2019). Neutropenia can result from defects in neutrophil production in the bone marrow, elevated apoptosis or from enhanced removal of activated neutrophils from the circulatory system (Duchene et al, 2017; Papadaki and Eliopoulos, 1998; Rankin, 2010).

Limited investigation of bone marrow in BTHS led to conflicting reports: hypocellularity and reduced myeloid maturation were detected in some BTHS patients without a complete block in

[1]School of Cellular and Molecular Medicine, Biomedical Sciences Building, University of Bristol, Bristol BS8 1TD, UK. [2]School of Biochemistry, Biomedical Sciences Building, University of Bristol, Bristol BS8 1TD, UK. [3]Cancer Research UK Scotland Institute, Glasgow G61 1BD, UK. [4]Bristol Dental School Research Laboratories, Dorothy Hodgkin Building, University of Bristol, Bristol BS1 3NY, UK. [5]These authors contributed equally: Przemysław Zakrzewski, Christopher M Rice. ✉E-mail: borko.amulic@bristol.ac.uk

neutrophil maturation (Barth et al, 1983; Rigaud et al, 2013; Steward et al, 2019), and short-term colony assays using hematopoietic stem cells purified from patient blood failed to detect any defects in neutrophil differentiation in vitro (Barth et al, 1983; Kuijpers et al, 2004). One study described elevated annexin V binding in circulating BTHS neutrophils, without increased apoptosis or phagocytic clearance by macrophages (Kuijpers et al, 2004). On the other hand, studies using shRNA-mediated *TAFAZZIN* knockdown in myeloid cell lines demonstrated increased caspase-3 activation, release of cytochrome c from mitochondria, and accelerated apoptosis in response to tafazzin deficiency (Makaryan et al, 2012). There is therefore no consensus on the mechanism of neutropenia in BTHS patients.

In mice, tafazzin deficiency causes embryonic lethality in C57BL/6 mice (Phoon et al, 2012; Wang et al, 2020), although other genetic backgrounds are viable. Immortalization of murine embryonic *Tafazzin*-KO myeloid progenitors demonstrated normal differentiation into mature neutrophils, normal mitochondrial electron transport chain assembly, and equivalent functional responses to wild-type (WT) cells (Sohn et al, 2022). The study did, however, detect slightly enhanced susceptibility to apoptosis upon Bcl-2 inhibition and subtle perturbations in expression of unfolded protein response (UPR) genes, raising the possibility that ER-mediated stress may affect viability of BTHS cells (Sohn et al, 2022).

To address the gap in understanding of BTHS neutropenia, we combined patient bone marrow analysis with our recently developed ex vivo neutrophil differentiation system (Naveh et al, 2024) to investigate the role of *TAFAZZIN* in neutrophil development from hematopoietic stem and progenitor cells (HSPC). We also performed a comprehensive analysis of neutrophil phenotype and function in BTHS patients, including proteomic analysis. Our work reveals a partial block in neutrophil differentiation and maturation in BTHS, accompanied by elevated intercellular calcium, hyperdegranulation, and evidence of enhanced NET formation. We detect alterations in the UPR pathway and propose that this may be the molecular basis of the observed defects in BTHS neutrophils.

# Results

## Partial impairment of bone marrow neutrophil maturation in BTHS

We examined neutrophil morphology in bone marrow aspirates of 5 BTHS patients (4–24 years old; all treated with G-CSF). The samples had similar features and were characterized by normocellularity, normal megakaryopoiesis and erythropoiesis, and unaffected eosinophil lineage (Figs. 1A and EV1A). As expected with G-CSF treatment, all samples exhibited marked 'left shift' i.e. an elevated number of promyelocyte precursors. However, there was a notable decrease in the number of metamyelocytes and mature neutrophils compared to reference values observed in healthy individuals; this was reflected in substantially reduced myeloid to erythroid ratio (Fig. 1B). These findings demonstrate a defect in neutrophil maturation in BTHS, as previously suggested (Barth et al, 1983; Rigaud et al, 2013; Steward et al, 2019).

## Impaired ex vivo terminal differentiation and maturation of BTHS neutrophils

To date, there have been no studies of maturation of tafazzin-deficient neutrophils using primary human progenitors. Our laboratory recently optimized a protocol to differentiate human neutrophils from CD34$^+$ HSPC isolated from a small amount of peripheral blood (Fig. 1C, adapted from (Naveh et al, 2024)). This protocol yields approximately 50% mature neutrophils, defined as CD66b$^+$CD15$^+$ cells. We used this system to compare neutrophil differentiation from CD34$^+$ progenitors of healthy control (HC) and BTHS patients, recruited from the NHS National Barth Syndrome Service at Bristol Royal Hospital for Children. We did not observe any difference in total cell number over 17 days of cell differentiation (Fig. 1D) or in proliferation ratio of BTHS and HC cells (fold change from day 3: HC = 90 ± 37 vs. BTHS = 82 ± 88) (Fig. 1E). Moreover, there was no increase in apoptosis in a subset of BTHS cultured neutrophils (Fig. EV1B), or any evident morphological changes (Fig. EV1C,D). However, flow cytometry (gating strategy in Fig. EV1E) uncovered an average 13.4% drop in the number of differentiated CD66b$^+$CD15$^+$ neutrophils in BTHS samples (HC = 48.08 ± 10.30% vs. BTHS = 34.69 ± 9.68%; *P* = 0.0145) (Fig. 1F). Consistently, the absolute number of BTHS neutrophils generated per thousand HSPC were reduced on average by 42.9% (HC = 54252 ± 50676 vs. BTHS = 31002 ± 43876; *P* = 0.3265) (Fig. EV1F). Moreover, flow cytometry revealed a 31% reduction in CD11b (HC = 1.00 ± 0.25 vs. BTHS = 0.69 ± 0.32; *P* = 0.0425) (Fig. 1G) and an 11% reduction in CD16 (HC = 1.00 ± 0.06 vs. BTHS = 0.89 ± 0.20; *P* = 0.1568) (Fig. 1H), which are commonly used neutrophil maturation markers. Importantly, this reduction was also reflected by an average 12.7% drop in the percentage of mature CD11b$^+$ neutrophils (HC = 37.08 ± 5.80% vs. BTHS = 24.40 ± 7.26%; *P* = 0.0013) and CD16$^+$ neutrophils (HC = 7.56 ± 3.43 vs. BTHS = 2.78 ± 2.01; *P* = 0.0029) at the end of differentiation (Fig. EV1G,H). In summary, BTHS patient HSPC show a moderate but consistent deficiency in neutrophil maturation.

To confirm a cell-intrinsic role for tafazzin in control of neutrophil differentiation, we performed shRNA knockdown experiments. First, to choose an optimal shRNA construct, we used the myeloid cell line PLB-985 to screen three vectors encoding *TAFAZZIN*-targeting shRNA. The most consistent results were obtained with shRNA 3, where we observed reductions in both tafazzin and CD11b in differentiated cells (Fig. EV2A,B). Next, we isolated CD34$^+$ HSPC from healthy donors and treated them with control non-targeting (n-t) or *TAFAZZIN*-targeting shRNA, on day 3 of differentiation (progenitors), which led to an ~45% reduction in tafazzin expression (Fig. EV2C,D). Similarly to BTHS patient cultured neutrophils, *TAFAZZIN* shRNA progenitors differentiated less efficiently than controls, as evidenced by a 30% reduction in CD11b (non-targeting shRNA = 1.00 ± 0.25 vs. *TAFAZZIN* shRNA = 0.70 ± 0.15; *P* = 0.0337) and a subtler 16% reduction in CD16 (non-targeting shRNA = 1.00 ± 0.52 vs. *TAFAZZIN* shRNA = 0.84 ± 0.67; *P* = 0.693) in differentiated neutrophils (Fig. 2A,B).

Finally, to test the effect of complete tafazzin deficiency, we used CRISPR/Cas9 to target exon 3 in CD34$^+$ HSPC (day 3 progenitors), with a non-targeting gRNA serving as control. Western analysis confirmed complete *TAFAZZIN* knockout at day 17 of

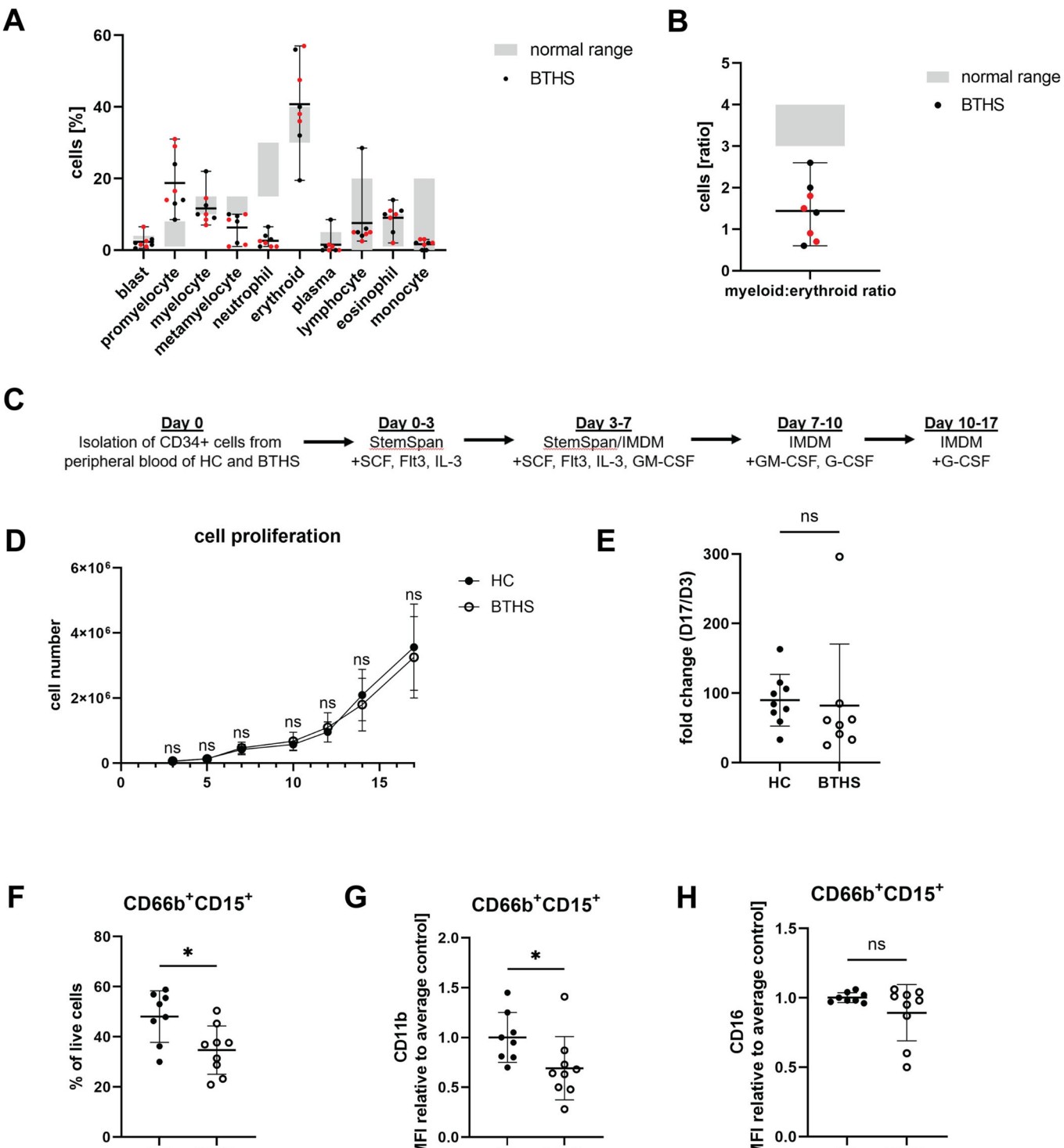

differentiation (Fig. 2C). Importantly, complete tafazzin deficiency resulted in approximately 50% reduction in cell proliferation, compared to non-targeting control, when comparing fold expansion between days 3 and 17 (non-targeting gRNA = 31 ± 15 vs. *TAFAZZIN* gRNA = 16 ± 8, *P* = 0.0154) (Figs. 2D and EV2E). Similarly to shRNA-treated neutrophils, *TAFAZZIN* knockout cells exhibited a ~ 18% decrease in the expression of differentiation

marker CD11b (non-targeting gRNA = 1.00 ± 0.06 vs. *TAFAZZIN* gRNA = 0.82 ± 0.20; *P* = 0.0131) (Fig. 2E) and ~16% decrease in the expression of CD16 (non-targeting gRNA = 1.00 ± 0.08 vs. *TAFAZZIN* gRNA = 0.85 ± 0.18; *P* = 0.026) (Fig. 2F). To assess changes in specific progenitor populations, we performed a time-course analysis of CD34, CD33 and CD64 expression and did not observe clear differences between tafazzin-deficient and control cells

Figure 1. Impaired terminal differentiation and maturation of BTHS neutrophils ex vivo.

(A) Differential count of bone marrow aspirates from BTHS patients stained with Wright-Giemsa, $n = 8$ (BTHS). The normal range (grey) represents values for healthy children. Samples from the same patient, obtained at different ages, are marked in red. (B) Myeloid:erythroid ratio for samples analyzed in (A), $n = 8$ (BTHS). Samples from the same patient, obtained at different ages, are marked in red. (C) Simplified overview of culture protocol for deriving neutrophils from HSPC. (D) Growth curve of ex vivo HSPC-derived neutrophil, $n = 9$ (HC), 8 (BTHS). (E) Fold change of total cell count of HSPC-derived neutrophils from day 3 to day 17 of differentiation, $n = 9$ (HC), 8 (BTHS). (F) Percent of CD66b$^+$CD15$^+$ neutrophils in HSPC-derived cells at the end of differentiation, $n = 8$ (HC), 9 (BTHS); $P = 0.0145$. (G) CD11b surface expression of CD66b$^+$CD15$^+$ HSPC-derived neutrophils at the end of differentiation, relative to averaged CD11b expression of control cells on a same day, $n = 8$ (HC), 9 (BTHS); $P = 0.0425$. (H) CD16 surface expression of CD66b$^+$CD15$^+$ HSPC-derived neutrophils at the end of differentiation, relative to averaged CD16 expression of control cells on a same day, $n = 8$ (HC), 9 (BTHS). Data information: Data are presented as mean ± SD. ns—not significant, *$P \leq 0.05$, assessed by unpaired $t$ test. Source data are available online for this figure.

(Fig. EV2F–H). These findings suggest that the observed drop in maturity markers, CD11b and CD16, may reflect a delay in differentiation rather than a block at a specific stage of neutrophil development. Taken together, experiments using HSPC-derived neutrophils demonstrate that tafazzin has a cell autonomous role in regulating neutrophil differentiation and maturation from progenitors.

## Immature status of circulating BTHS neutrophils

We next examined circulating neutrophils in BTHS patients ($n = 15$, 4 out of 15 not receiving G-CSF therapy) and healthy controls ($n = 15$, all untreated). Whole blood flow cytometric immunophenotyping, using previously described gating strategies (Rice et al, 2023), revealed reduced levels of the maturity markers CD10 (HC = 4036 ± 2343 vs. BTHS = 2168 ± 1969; $P = 0.0253$) and CD16 (HC = 20222 ± 10030 vs. BTHS = 14037 ± 9303; $P = 0.0908$) (Fig. 3A). We did not observe any differences in CD101 and CD62L between HC and BTHS patients (Fig. EV3A). Both CD10 and CD16 expression were normal in G-CSF untreated non-neutropenic BTHS patients ($n = 4$) (Fig. EV3B), suggesting that their hematopoietic compartment is not affected. G-CSF mobilizes immature neutrophils from the bone marrow, resulting in accumulation of circulating neutrophils with reduced CD10 and CD16 (Marini et al, 2017), making it difficult to conclude whether the observed immaturity is due to G-CSF treatment or *TAFAZZIN* mutation (as shown in HSPC differentiation experiments).

## BTHS neutrophils exhibit elevated degranulation and competent bacterial killing

We next used flow cytometry to examine neutrophil surface markers associated with activation. Interestingly, the secondary granule marker CD66b expression was elevated on patient neutrophils compared to controls (HC = 2385 ± 401 vs. BTHS = 4125 ± 1156; $P < 0.0001$), suggesting increased spontaneous in vivo exocytosis of secondary granules (Fig. 3B). This finding was independent of G-CSF therapy (Fig. EV3C). To test if elevated degranulation of secondary granules occurs in response to live bacteria, we stimulated purified neutrophils with *Streptococcus pyogenes* and quantified exposure of CD66b over 40 min of co-incubation (Fig. 3C). Interestingly, we found elevated degranulation in naive BTHS neutrophils ($t = 0$), confirming our in vivo findings on increased spontaneous degranulation (Fig. 3B). Upon bacterial stimulation, the fold increase in surface CD66b was comparable between BTHS and HC neutrophils (Fig. EV3D). Next, we assessed degranulation of primary granules by measuring surface exposure

of CD63 on isolated neutrophils, with or without stimulation with the bacterial peptide N-formylmethionine-leucyl-phenylalanine (fMLP). There was no difference in CD63 exposure in naive neutrophils (Fig. 3D). Upon fMLP stimulation, however, BTHS neutrophils ($n = 31$) demonstrated more than 1.7-fold increase in activation compared to HC ($n = 29$); (HC = 1213 ± 700 vs. BTHS = 2115 ± 1927; $P = 0.0032$), indicating increased degranulation of primary granules in patient cells (Fig. 3D). Finally, we quantified plasma abundance of MPO, a primary granule protein, and found elevated circulating MPO levels in BTHS patients compared to healthy control plasma samples (Fig. 3E). In summary, neutrophils from BTHS patients are more prone to spontaneous and induced degranulation of primary and secondary granules.

A previous report demonstrated elevated rates of phosphatidyl serine (PS) externalization in BTHS neutrophils, although much lower than typically associated with neutrophil apoptosis (Kuijpers et al, 2004). We stained a portion of our neutrophils with annexin V to quantify PS surface exposure. As previously reported, we observed increased PS exposure in naive neutrophils (HC = 3.80 ± 1.48% vs. BTHS = 6.31 ± 2.02%; $P = 0.0344$) (Fig. 3F). We speculated that elevated PS exposure in BTHS neutrophils results from increased degranulation. To test this, we demonstrated that fMLP stimulation increases annexin V positivity and equalized expression in HC and BTHS neutrophils (HC = 17.86 ± 12.88% vs. BTHS = 14.36 ± 5.19%; $P = 0.550$) (Fig. 3G). In summary, elevated PS exposure of BTHS neutrophils is caused by increased degranulation and not apoptosis.

Secondary granules contain multiple cytoadhesive molecules and augmented degranulation might result in excessive endothelial interactions, leading to sequestration of neutrophils in different organs. We therefore analyzed spleen and liver histological sections from WT and *Tafazzin*$^{-/-}$ mice, for neutrophil abundance, using a neutrophil-specific calgranulin antibody (Fig. EV3E). We found no difference in the number of organ-resident neutrophils in these tissues under steady state conditions. Interestingly, in contrast to what is observed in BTHS patients, *Tafazzin*-KO mice, on FVB (sensitive to the Friend leukemia virus) genetic background, had equivalent absolute neutrophil counts to those of WT mice (Fig. EV3F), suggesting that tafazzin may have disparate roles in neutrophil development in mice and humans.

Next, we investigated whether neutrophils from BTHS patients show other functional alterations. Using a luminol based assay, we quantified ROS production in response to phorbol 12-myristate 13-acetate (PMA) and concanavalin A (con A) and found no differences compared to HC (Fig. 3H). Moreover, we found no difference in NET formation with the strong soluble inducer PMA (Fig. 3I). Moreover, BTHS neutrophils phagocytosed pHrodo$^{TM}$

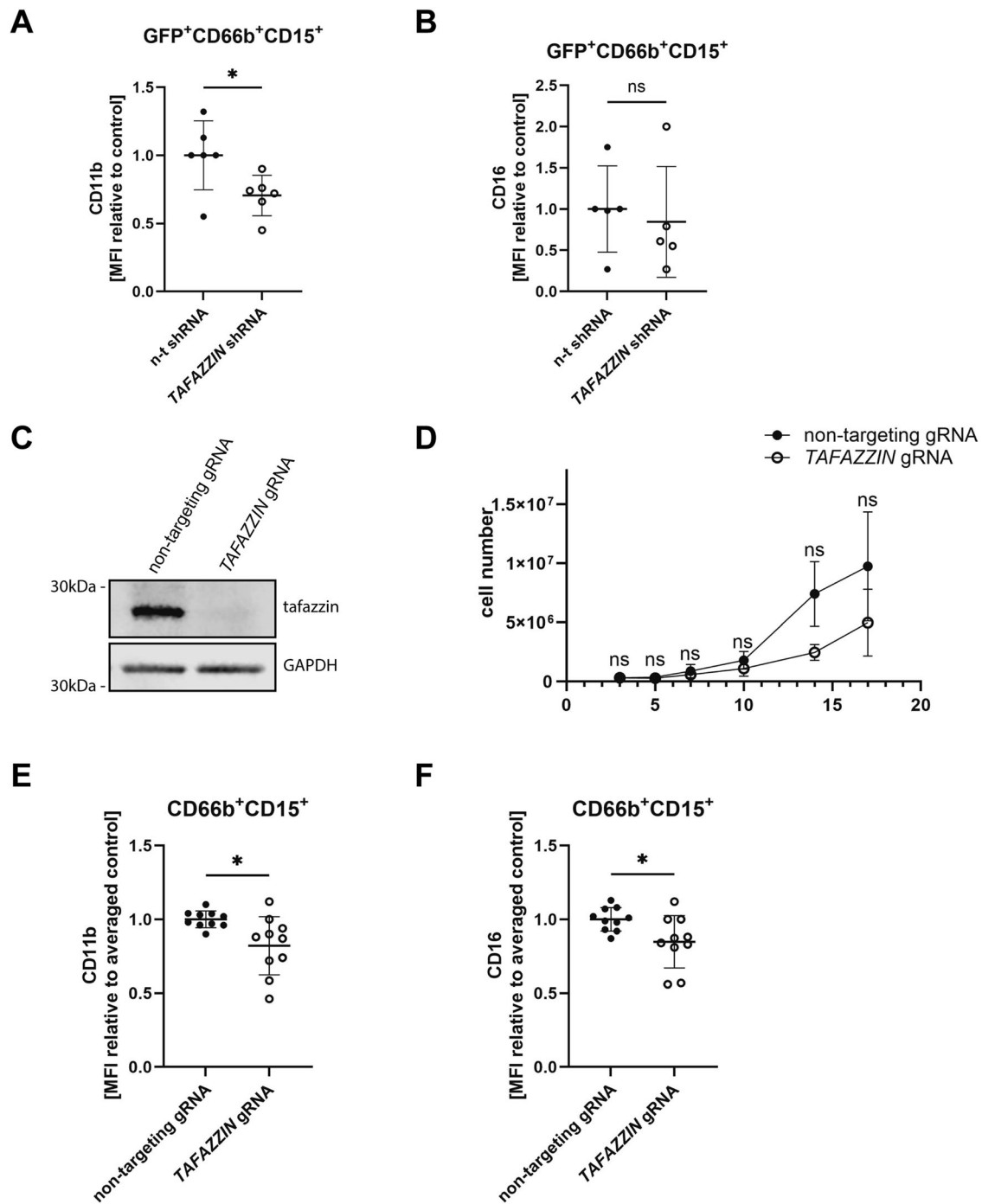

**Figure 2. Tafazzin regulates neutrophil development.**

(**A**) CD11b surface expression of control or tafazzin shRNA-treated CD66b+CD15+GFP+ HSPC-derived neutrophils at the end of differentiation, relative to non-targeting (n-t) shRNA control, $n = 6$ (biological repeats); $P = 0.0337$. (**B**) CD16 surface expression of control or tafazzin shRNA-treated CD66b+CD15+GFP+ HSPC-derived neutrophils at the end of differentiation, relative to n-t shRNA control, $n = 5$ (biological repeats). (**C**) Representative tafazzin western blot of CRISPR/Cas9-edited HSPC-derived neutrophils at the end of differentiation. (**D**) Growth curve of CRISPR/Cas9-edited HSPC-derived neutrophils, $n = 4$ (biological repeats). (**E**) CD11b surface expression of CD66b+CD15+ CRISPR/Cas9-edited HSPC-derived neutrophils at the end of differentiation, relative to non-targeting gRNA control, $n = 10$ (biological repeats); $P = 0.0131$. (**F**) CD16 surface expression of CD66b+CD15+ CRISPR/Cas9-edited HSPC-derived neutrophils at the end of differentiation, relative to non-targeting gRNA control, $n = 10$ (biological repeats); $P = 0.026$. Data information: Data are presented as mean ± SD. ns—not significant, *$P \leq 0.05$, assessed by unpaired $t$ test. Source data are available online for this figure.

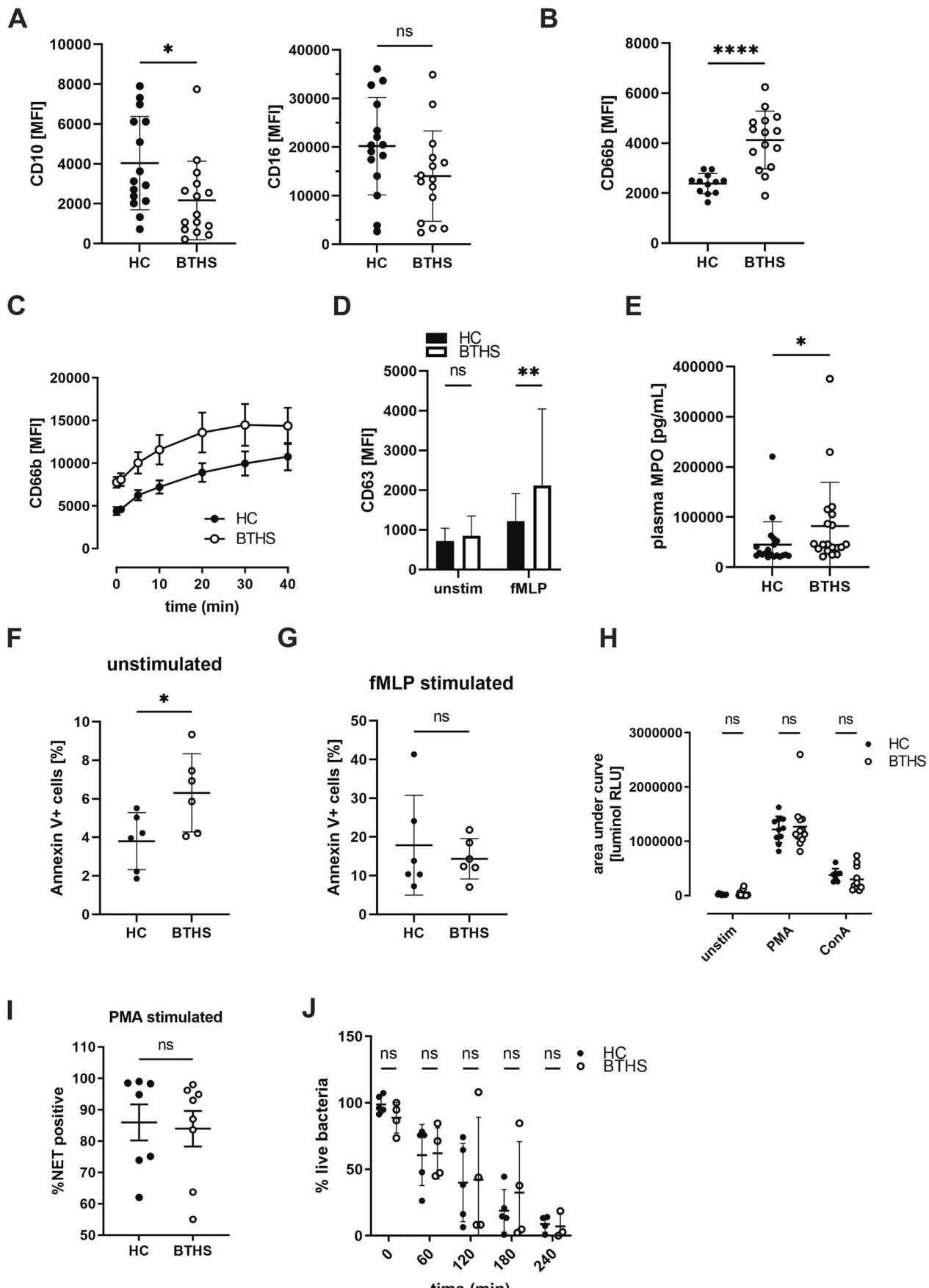

Figure 3. Characterization of peripheral blood neutrophils from BTHS patients.

(A) Surface expression of CD10 (left) and CD16 (right) in circulating neutrophils, $n = 15$ (HC, BTHS); $P$ (CD10) = 0.0253. (B) Surface expression of CD66b in circulating neutrophils, $n = 12$ (HC), 15 (BTHS); $P < 0.0001$. (C) Time-course of CD66b surface expression in isolated neutrophils stimulated with *Streptococcus pyogenes*, MOI = 100, $n = 9$ (HC), 7 (BTHS). (D) Surface expression of CD63 in unstimulated (unstim) and fMLP-stimulated (300 nM) isolated neutrophils, $n$ (unstim) = 29 (HC, BTHS), $n$ (fMLP) = 31 (HC, BTHS); $P = 0.0032$. (E) MPO concentration in plasma, $n = 20$ (HC), 19 (BTHS); $P = 0.0164$. (F) Percentage of annexin V+ unstimulated neutrophils, $n = 6$ (HC, BTHS); $P = 0.0344$. (G) Percentage of annexin V+ neutrophils, after stimulation with 300 nM fMLP, $n = 6$ (HC, BTHS). (H) Area under the curve (AUC) quantification of ROS, detected by luminol, $n$ (unstim, PMA) = 10 (HC), 12 (BTHS), $n$ (ConA) = 8 (HC), 10 (BTHS). (I) NET release in response to 50 nM PMA, $n = 7$ (HC), 8 (BTHS). (J) Quantification of viable *Streptococcus pyogenes* bacteria relative to "serum only" control, after incubation with isolated neutrophils, $n = 4$ (HC, BTHS). Data information: Data are presented as mean ± SD. ns—not significant, *$P \leq 0.05$, **$P \leq 0.01$, ****$P \leq 0.0001$, assessed by unpaired $t$ test (A, B, E–G, I, J) and two-way ANOVA (D, H). Source data are available online for this figure.

*Streptococcus pyogenes* at similar rates to HC neutrophils (Fig. EV3G). Neutrophils from BTHS patients ($n = 4$) and HC ($n = 4$) also demonstrated equivalent killing rates of opsonized *Streptococcus pyogenes* bacteria (Fig. 3J).

In summary, BTHS patient neutrophils display enhanced secondary and primary degranulation, normal PMA-induced NETosis and maintain effective killing of a Gram-positive pathogen.

## Mitochondrial function is not affected in BTHS patient neutrophils

Mitochondria in BTHS patient neutrophils have not previously been studied. We found increased mitochondrial abundance in patients relative to control, quantified with MitoTracker™ dye (HC = 1.00 ± 0.22 vs. BTHS = 1.66 ± 0.52; $P = 0.001$) (Fig. 4A). Interestingly, BTHS patient neutrophils had reduced mitochondrial membrane potential, as evidenced by reduction in tetramethylrhodamine ethyl ester (TMRE) fluorescence compared to HC neutrophils (HC = 31 ± 19% TMRE low cells vs. BTHS = 47 ± 25% TMRE low cells; $P = 0.0349$) (Fig. 4B), suggesting that the increased abundance may be a compensatory mechanism for impaired activity. To test this hypothesis, we used Seahorse mitochondrial stress analysis to test mitochondrial respiration in isolated BTHS primary neutrophils ($n = 6$, with 3 out of 6 patients treated with G-CSF) and HC. While no statistically significant differences were observed in mitochondrial ATP production, basal respiration, maximal respiration, or spare respiratory capacity (Fig. 3C), stratifying the data revealed that the increased mitochondrial activity was observed in G-CSF-treated BTHS patients (Fig. EV3A). This supports previous reports indicating that G-CSF treatment leads to mobilization of immature neutrophils that have increased mitochondria content (Marini et al, 2017; Naveh et al, 2024; Rice et al, 2018). Therefore, the mitochondrial phenotypes we observed in BTHS neutrophils can likely be attributed to the effects of G-CSF therapy rather than intrinsic defects caused by tafazzin deficiency.

Finally, as several studies reported elevated mitochondrial ROS (mtROS) in BTHS cardiomyocytes (Liu et al, 2021; Saric et al, 2015), we quantified mtROS in BTHS neutrophils using a MitoSOX™ flow cytometry assay. The percentage of MitoSOX-positive cells (compared to unstained samples) (Fig. EV4B) was elevated in BTHS samples, both in homeostatic conditions (HC = 19.20 ± 15.72 vs. BTHS = 31.85 ± 16.66; $P = 0.058$) (Fig. 4D) and after stimulation with fMLP (HC = 11.27 ± 5.35% vs. BTHS = 30.79 ± 17.46%; $P = 0.0084$) (Fig. 4E), although this difference was not observed when analyzing total MitoSOX MFI (Fig. EV4C,D). This suggests that more BTHS cells produced detectable levels of

mtROS, whereas the overall intensity per cell may be lower, potentially indicating a more widespread but less intense ROS production across the BTHS population. Moreover, we did not detect any mtROS differences in BTHS HSPC-derived neutrophils in a limited number of samples (Fig. EV4E,F). In conclusion, mitochondria in GCSF-treated BTHS patients are characterized by elevated abundance, reduced membrane potential and elevated mtROS. However, we failed to detect any evident impairment of overall mitochondrial activity.

## Tafazzin regulates primary degranulation and calcium-induced NETosis

Excessive mtROS can affect calcium homeostasis, which in turn can regulate neutrophil activation (Feno et al, 2019). As we observed increased mtROS in BTHS neutrophils, we decided to quantify intracellular calcium level in these neutrophils using X-Rhod-1, a cell permeant dye that exhibits increased fluorescence after binding $Ca^{2+}$. We found a >twofold increase in intracellular calcium level of circulating BTHS cells, compared to HC neutrophils (HC = 1.00 ± 0.08 vs. BTHS = 2.33 ± 1.26; $P = 0.0025$) (Fig. 5A). To determine whether this change is caused by G-CSF therapy or a cell-intrinsic effect of tafazzin deficiency, we analyzed calcium levels in HSPC-derived neutrophils and observed an almost threefold increase in X-Rhod-1 signal in patient stem cell-derived neutrophils, compared to HC (HC = 2570 ± 671 vs. BTHS = 5662 ± 1160; $P = 0.0162$) (Fig. 5B).

Calcium is an essential activator of neutrophil inflammatory responses (Hann et al, 2020), prompting us to ask whether tafazzin-deficient neutrophils have altered functional responses. To test this, we utilized the calcium ionophore A23187, which induces calcium influx across the plasma membrane and strongly activates neutrophils (Kenny et al, 2017). As expected, A23187 induced strong degranulation of neutrophil elastase and MPO measured by ELISA (Fig. 5C,D). We observed a 1.7-fold increase in degranulation of neutrophil elastase (non-targeting gRNA = 4251.81 ± 971.07 pg/mL vs. *TAFAZZIN* gRNA = 7175.88 ± 411.93 pg/mL; $P = 0.0058$) and a 1.8-fold increase in MPO degranulation (non-targeting gRNA = 49281.99 ± 11404.44 pg/mL vs. *TAFAZZIN* gRNA = 86918.78 ± 41301.97 pg/mL; $P = 0.1107$).

We next investigated NET formation in *TAFAZZIN* knockout HSPC-derived neutrophils in response to calcium-independent (PMA) and calcium-dependent stimuli (A23187), using a live cell imaging assay. There was no difference in chromatin release in response to PMA in a live cell imaging assay with SYTOX green DNA dye (Fig. 5E). On the other hand, A23187-induced NETosis was significantly increased in tafazzin-deficient neutrophils compared to control cells (non-targeting gRNA = 3006947 ± 633265 vs.

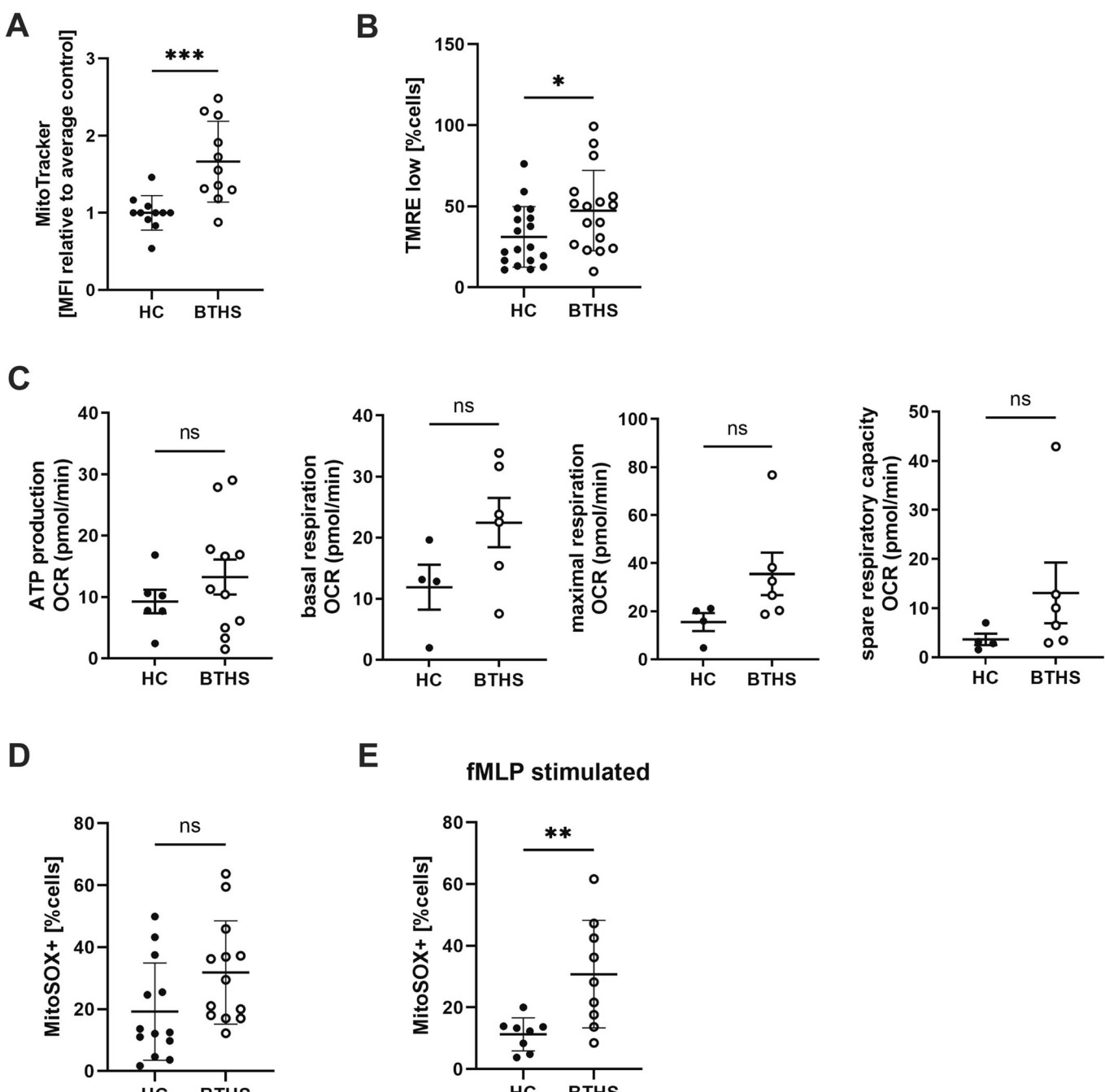

**Figure 4. Mitochondrial perturbations in BTHS patient circulating neutrophils.**

(A) Average MitoTracker median fluorescence of isolated BTHS patient circulating neutrophils normalized to HC (on the same day), $n = 11$ (HC, BTHS); $P = 0.001$. (B) Average percentage of neutrophils with low TMRE signal, $n = 18$ (HC), 17 (BTHS); $P = 0.0349$. (C) Average rates of ATP production, basal respiration, maximal respiration, and spare respiratory capacity, measured by Seahorse analyzer, $n$ (ATP) = 6 (HC), 11 (BTHS), $n$ (basal, maximal, spare respiration) = 4 (HC), 6 (BTHS). (D) Percentage of MitoSOX-positive neutrophils $n = 13$ (HC, BTHS). (E) Percentage of MitoSOX-positive neutrophils, after stimulation with 300 nM fMLP, $n = 8$ (HC), 9 (BTHS); $P = 0.0084$. Data information: Data are presented as mean ± SD. ns—not significant, *$P \leq 0.05$, **$P \leq 0.01$, ***$P \leq 0.001$, assessed by unpaired t-test. Source data are available online for this figure.

*TAFAZZIN* gRNA = 4216884 ± 1015170; $P = 0.0115$) (Fig. 5F). To confirm that cell death detected with SYTOX green was bona fide NETosis, rather than apoptosis or necrosis, we stained the cells with NET-specific antibody 3D9 (Tilley et al, 2022), and found similarly elevated production of NETs in tafazzin CRISPR KO cells (non-targeting gRNA = 52.00 ± 4.68% vs. *TAFAZZIN* gRNA = 62.60 ± 2.95%; $P = 0.0120$) (Fig. 5G,H). In conclusion, both circulating and stem cell derived BTHS neutrophils exhibit elevated intracellular

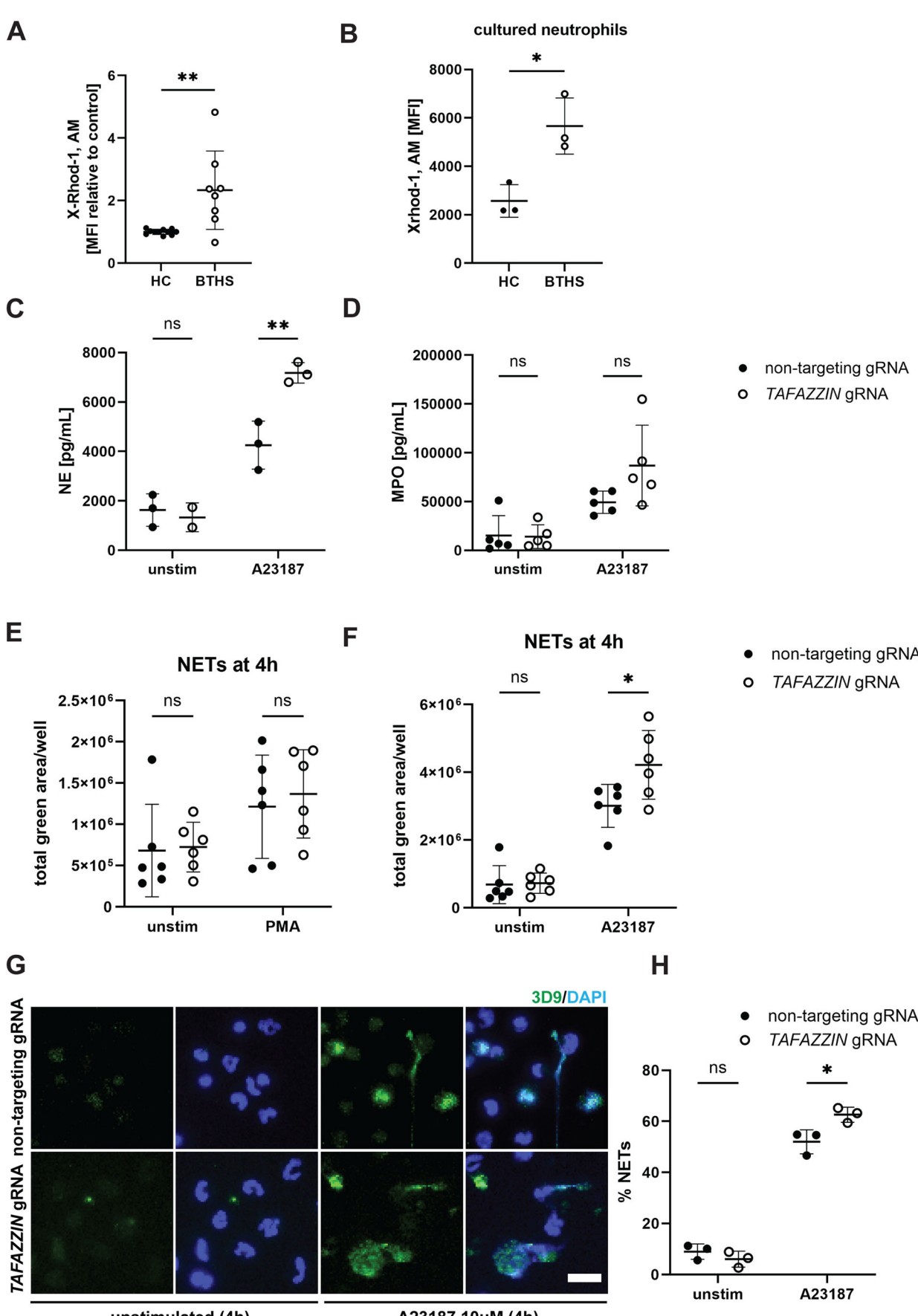

◀  **Figure 5.  Calcium homeostasis in BTHS neutrophils.**

(A) Average median fluorescence of calcium dye X-Rhod-1, normalized to HC (on the same day), $n = 11$ (HC), 8 (BTHS); $P = 0.0025$. (B) Quantification of average median fluorescence of X-Rhod-1 in HSPC-derived BTHS neutrophils (D17), $n = 3$ (HC, BTHS); $P = 0.0162$. (C) NE concentration in medium released by CRISPR/Cas9-edited HSPC-derived neutrophils before and after stimulation with A23187 (2.5 µM) at 30 min post-stimulation, $n = 3$ (biological repeats); $P = 0.0058$. (D) MPO concentration in medium released by CRISPR/Cas9-edited HSPC-derived neutrophils before and after stimulation with A23187 (2.5 µM) at 30 min post-stimulation, $n = 5$ (biological repeats). (E) Quantification of NET release by CRISPR/Cas9-edited HSPC-derived neutrophils before or after stimulation with PMA (50 nM) at 4 h post-stimulation, $n = 6$ (biological repeats). (F) Calculation of NET release by CRISPR/Cas9-edited HSPC-derived neutrophils before or after stimulation with A23187 (10 µM) at 4 h post-stimulation, $n = 6$ (biological repeats); $P = 0.0115$. (G) Representative epifluorescence images showing anti-3D9 (green) and DAPI (blue) staining of CRSIPR/Cas9-edited HSPC-derived neutrophils at 4 h after stimulation with A23187 (10 µM) or not stimulated, scale bar = 20 µm. (H) Percentage of 3D9-positive cells after 4 h stimulation with A23187 (10 µM), $n = 3$ (biological repeats, at least 100 cells were counted in randomized fields of view); $P = 0.0120$. Data information: Data are presented as mean ± SD. ns—not significant, $^{*}P ≤ 0.05$, $^{**}P ≤ 0.01$, assessed by unpaired $t$ test (A, B, H) and two-way ANOVA (C–F). Source data are available online for this figure.

calcium concentration, which promotes higher rates of degranulation and calcium-dependent NET formation.

## Elevated UPR signaling in BTHS patients

To better understand developmental changes in BTHS patient neutrophils, we performed tandem mass tag (TMT) proteomic analysis of circulating neutrophils from patients ($n = 4$ G-CSF-treated and $n = 1$ untreated, non-neutropenic patient), as well as healthy controls ($n = 5$). We detected a total of 4056 proteins (Fig. EV5A), including 306 proteins with significantly altered expression ($P < 0.05$), 91 of which had >twofold difference. Ingenuity Pathway Analysis (IPA) revealed multiple significantly up- and downregulated pathways in BTHS neutrophils (Fig. 6A). Notably, the most strongly enriched pathway in BTHS was oxidative phosphorylation (Fig. EV5B), closely followed by fatty acid oxidation (Fig. EV5C), consistent with elevated mitochondrial load, as detected with MitoTracker™ (Fig. 4A). Intriguingly, the second most enriched pathway in BTHS neutrophils was the unfolded protein response (UPR), which was previously implicated in neutrophil dysregulation in *Tafazzin*-KO mice (Sohn et al, 2022). Specifically, we observed an upregulation of eight UPR proteins (Fig. 6B) and four proteins associated with mitochondrial UPR (mt-UPR, Fig. 6C). These changes appeared to be GCSF- and neutropenia-dependent, as the only untreated, non-neutropenic patient in our study consistently exhibited less pronounced alteration (sample *BTHS5*). Of note, changes in the expression of mt-UPR proteins were more uniform among patients (Fig. EV5D).

To confirm these changes, we quantified UPR protein abundance by western blot (Fig. 6D). To avoid confounding by G-CSF treatment, we used HSPC-derived neutrophils from BTHS patients and HC, which are exposed to equivalent amounts of G-CSF over the 10 days of the differentiation protocol. We observed a trend for increased abundance of IRE1α, BiP and p-eIF2α and a significant upregulation of HSP60 (Fig. EV5E), supporting the proteomic results.

One of the key upstream regulators of UPR signaling that plays a critical role in cell function is PERK (Hetz et al, 2020). To investigate its involvement in neutrophil differentiation from HSPCs, we first assessed the expression and activity of PERK throughout the differentiation process. Western blot analysis revealed that both PERK expression and activity, measured by the phosphorylation of its downstream effector eIF2α, were abundant in HSPC and myeloid progenitors, and progressively decreased from day 7 (D7) to differentiation endpoint (D17)

(Fig. 6E). Next, we used GSK2606414, a small molecule inhibitor targeting the ATP binding site, to inhibit its activity. This partially blocked neutrophil differentiation, as evidenced by the reduced number of CD66b$^{+}$CD15$^{+}$ (DMSO = 51.12 ± 15.79% vs. PERKi = 39.02 ± 18.92%; $P = 0.043$) and CD11b$^{+}$ (DMSO = 39.43 ± 14.19% vs. PERKi = 24.58 ± 14.56%; $P = 0.111$) neutrophils (Fig. 6F), reduced CD11b surface expression (DMSO = 1.00 ± 0.44 vs. PERKi = 0.77 ± 0.38, $P = 0.254$) (Fig. EV5F), and diminished proliferation by the end of the differentiation process (DMSO = 27 ± 16 vs. PERKi = 9 ± 8; $P = 0.0064$) (Fig. EV5G). In a complementary approach, we used the PERK activator CCT020312 to increase the activity of this signaling pathway. PERK activation had little effect on the number of CD66b$^{+}$CD15$^{+}$ neutrophils obtained in the differentiation protocol (DMSO = 56.17 ± 11.43% vs. PERKa = 49.75 ± 7.04%; $P = 0.585$) or the percentage of CD11b$^{+}$ cells (DMSO = 45.82 ± 15.27% vs. PERKa = 36.47 ± 11.94%; $P = 0.334$) (Fig. EV5H), but it did reduce CD11b surface expression (DMSO = 1.00 ± 0.14 vs. PERKa = 0.71 ± 0.14, $P = 0.0054$) (Fig. EV5I). We also observed a trend towards diminished proliferation by the end of the differentiation process (DMSO = 43 ± 8 vs. PERKa = 36 ± 11; $P = 0.1659$) (Fig. EV5J). Although not all results reached threshold of $P ≤ 0.05$ for statistical significance, the consistent downward patterns observed across these parameters suggest that dysregulation of PERK activity may negatively impact neutrophil maturation and argue for additional studies of UPR in BTHS neutropenia.

## Discussion

Despite the prominence of neutropenia in BTHS patients, our understanding of the molecular and physiological causes of this phenomenon has been limited. Our study reveals that tafazzin has a cell-intrinsic role in regulating neutrophil differentiation and confirms a partial delay in granulopoiesis in BTHS patients. We also show that tafazzin-deficient neutrophils exhibit enhanced degranulation and elevated rate of NETosis. Finally, we confirm alterations in UPR signaling in BTHS neutrophils, which might be a contributing factor to the observed developmental and functional changes.

We observed a decrease in the number of mature neutrophils in BTHS bone marrow samples. This is consistent with the original bone marrow examination by Barth (Barth et al, 1983) ($n = 2$) and subsequent work (Rigaud et al, 2013; Steward et al, 2019) that demonstrated hypocellularity in some, but not all, BTHS patients. It remains unclear whether reduction in bone marrow neutrophils

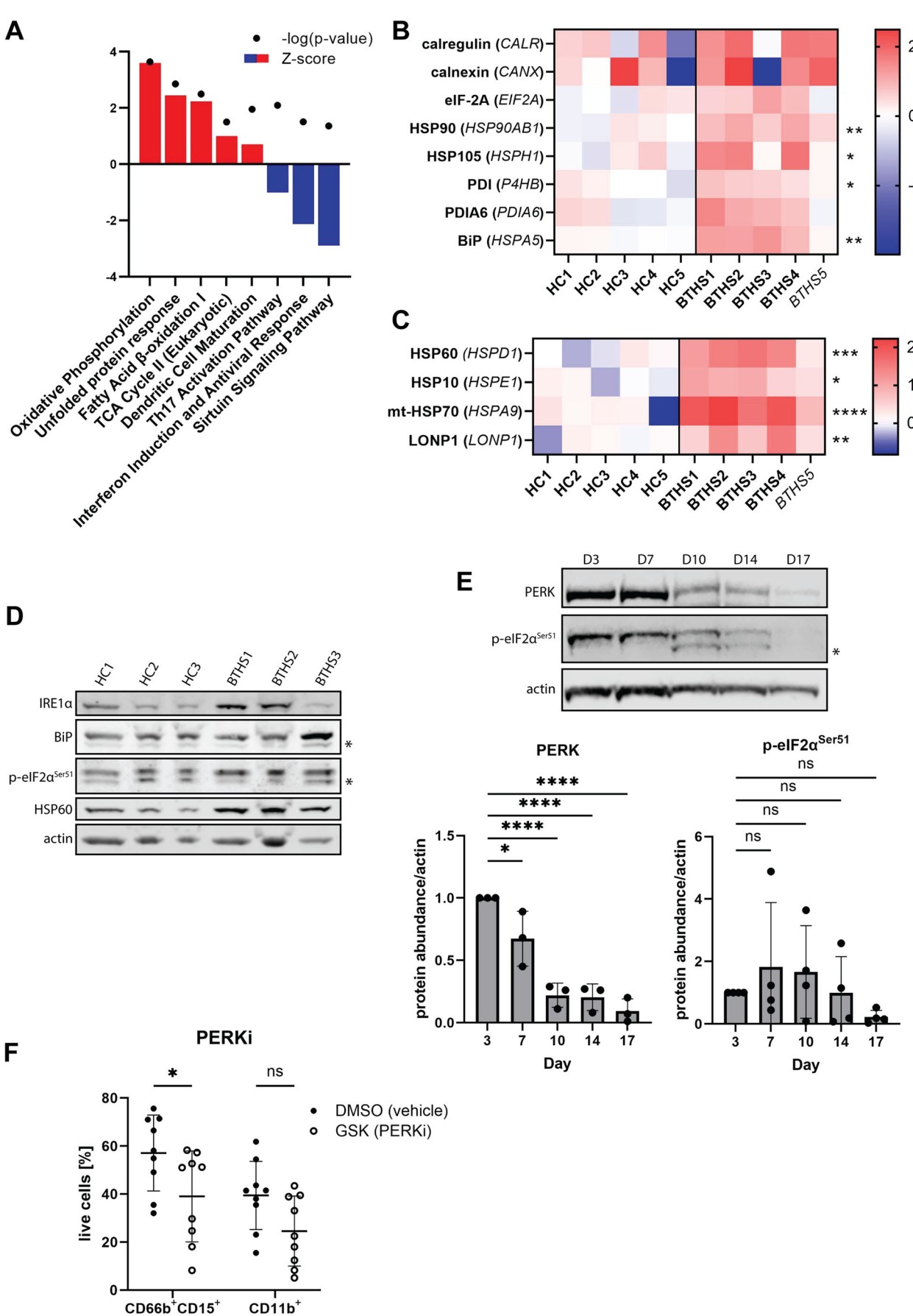

◀ **Figure 6. Elevated UPR signaling in BTHS neutrophils.**

(A) Plot showing significantly upregulated (positive Z-score, red) and downregulated (negative Z-score, blue) IPA canonical pathways in circulating neutrophils from BTHS patients; $n = 5$ (HC), 5 (BTHS including $n = 4$ G-CSF-treated and $n = 1$ untreated, non-neutropenic patient). (B) Heat map depicting upregulated UPR-related proteins in circulating BTHS neutrophils, displayed as log2 fold change over averaged HC. (C) Heat map depicting upregulated mitochondrial UPR-related proteins in circulating BTHS neutrophils, displayed as log2 fold change over averaged HC. (B, C) BTHS5 - non-neutropenic BTHS patient. (D) Western blot of selected UPR-related proteins in HC or BTHS patient HSPC-derived neutrophils at the end of differentiation (D17), $n = 3$ (HC, BTHS); asterisks indicate nonspecific bands. (E) Western blot depicting the level of PERK expression and eIF2α phosphorylation during differentiation of HSPC-derived neutrophils and protein abundance quantification normalized to actin, $n = 3$ (PERK), 4 (p-eIF2α); asterisk indicate unspecific band. $P$ (day 7) $= 0.0306$, $P$ (day 10–17) $< 0.0001$. (F) Average percentage of HSPC-derived neutrophils (CD66b$^+$CD15$^+$) and mature neutrophils (CD66b$^+$CD15$^+$CD11b$^+$) at D17, after treatment with PERK inhibitor (GSK2606414; 1 μM, added on day 3, 5, 7, 10, and 14 of culture) or vehicle control (DMSO), $n = 9$ (DMSO, PERKi); $P = 0.0439$. Data information: Data are presented as mean ± SD. ns—not significant, $*P \le 0.05$, $**P \le 0.01$, $***P \le 0.001$, $****P \le 0.0001$, assessed by Welch's $t$ test after log$_2$ transformation (A–C), unpaired $t$ test (E) and two-way ANOVA (F). Source data are available online for this figure.

exists in a subset of patients and whether this is defined by the nature of the tafazzin mutation or by extrinsic factors. The maturation deficiency largely manifests in later stages of neutrophil development, affecting metamyelocytes and band neutrophils. How this leads to neutropenia, rather than simply a decrease in maturity of circulating neutrophils, remains unexplained, and implies the involvement of additional extrinsic factors.

We did not observe any evident hallmarks of early apoptosis in BTHS cultured neutrophils, which is reflected in largely unaltered proliferation rates. We showed that primary BTHS neutrophils exhibited a modest increase in annexin V binding, which is also induced by fMLP stimulation. Consistent with the conclusion of Kuijpers et al, (Kuijpers et al, 2004) that increased binding of annexin V does not correlate with apoptosis, we propose that elevated PS in BTHS neutrophils results from increased neutrophil degranulation, which is known to alter surface lipid composition and transiently expose PS (Frasch et al, 2008; Frasch et al, 2004). We conclude that neutropenia in BTHS patients is most likely not caused by accelerated apoptosis.

Intriguingly, we also observed increased functional responses in tafazzin-deficient neutrophils. Both spontaneous and bacterial-induced degranulation were elevated in patient neutrophils, while enhanced released of primary granule components was observed in tafazzin CRISPR KO cultured neutrophils, suggesting a cell intrinsic role for transacylation of CL in regulation granule release.

*TAFAZZIN* knockout neutrophils also show elevated rates of inflammatory NETotic cell death in response to calcium mobilization, although this was not tested in circulating BTHS patient neutrophils. Increased NET formation may lead to peripheral depletion of neutrophils, possibly contributing to neutropenia. NET release is also a potent inflammatory signal that activates macrophages and other immune cells (Apel et al, 2021; Knackstedt et al, 2019), potentially lowering the threshold for acute and chronic inflammation (Castanheira and Kubes, 2019; Wigerblad and Kaplan, 2023). These findings highlight a major knowledge gap on inflammatory processes in BTHS and suggest that inflammation should be further investigated as an etiological factor in BTHS.

Several mechanisms can be proposed for enhanced granule exocytosis and NETosis in tafazzin-deficient neutrophils. First, we observed elevated levels of intracellular calcium in BTHS cultured and primary neutrophils; Ca$^{2+}$ flux promotes neutrophil degranulation (Hann et al, 2020; Liu et al, 2021) and NET release (Kenny et al, 2017). Mitochondria are known to regulate intracellular calcium stores (Romero-Garcia and Prado-Garcia, 2019) and this

mechanism may be perturbed by alterations in cardiolipin composition (Ghosh et al, 2020). Secondly, it is known that ROS production, including mtROS, can trigger the disassembly of filamentous actin (F-actin) (Sakai et al, 2012; Vorobjeva et al, 2017; Yadav et al, 2022), which is needed for both mobilization of granules (Jog et al, 2007; Mitchell et al, 2008) and NET formation (Metzler et al, 2014). We observed elevated mtROS in BTHS neutrophils, suggesting that activation in BTHS may be facilitated by ROS-induced actin degradation. Finally, degranulation is regulated by UPR sensors, which we found to be dysregulated in both primary and cultured BTHS neutrophils. In a model of acute lung injury, ER stress triggered by the IRE1/XBP1 pathway promotes degranulation, and specific depletion of XBP1 in neutrophils reduces granule secretion (Hu et al, 2015). Similarly, in lupus, neutrophils exhibit elevated IRE1α activity that was correlated with increased extracellular elastase activity (Sule et al, 2021). These findings imply that fluctuations in UPR during the physiological turnover of neutrophils could influence their inflammatory response.

In contrast to previous findings associating tafazzin deficiency with disruptions in mitochondrial respiration (Dudek et al, 2013; Hsu et al, 2015; Kagan et al, 2023; Lou et al, 2018), we found no defects in the basal and maximal respiratory rates nor in the rate of ATP production in circulating BTHS neutrophils, although the interpretation of these findings is complicated by the fact that G-CSF treatment mobilizes immature neutrophils that have elevated mitochondrial content and activity (Marini et al, 2017; Naveh et al, 2024; Rice et al, 2018). Indeed, when we stratified our Seahorse data, we observed that changes in mitochondrial activity were exclusively present in BTHS patients receiving G-CSF treatment, suggesting that these changes are more likely attributed to the treatment rather than intrinsic mitochondrial alternations caused by tafazzin dysfunction. Neutrophils differ from other cells in that they have fewer mitochondria per cell, which has traditionally led to the belief that these cells primarily rely on glycolysis and are not dependent on active mitochondria; however, recent advancements have highlighted the significance of these organelles in neutrophil development and function (Cao et al, 2022; Vorobjeva et al, 2023). Future studies should carefully examine mitochondrial metabolism in BTHS progenitors and delineate the specific contributions of G-CSF treatment and tafazzin deficiency to neutrophil mitochondrial dynamics and functionality.

Interestingly, our proteome analysis uncovered alternations in UPR and mtUPR pathways in BTHS neutrophils, which is in

accordance with findings from *Tafazzin*-KO mice (Sohn et al, 2022). While UPR pathways are traditionally associated with protein misfolding in the ER, recent research has highlighted their critical roles in regulating immunity and inflammation (Grootjans et al, 2016). For instance, experiments using HL-60 cells demonstrated that in vitro neutrophil differentiation relies on the stage-specific expression of canonical UPR regulators (Tanimura et al, 2018). Notably, inhibition of these three proteins reduced the expression of CD11b and morphological differentiation of those cells. Our data confirm these reports, showing that PERK activity is primarily confined to early stages of neutrophil differentiation, as both the expression of PERK and phosphorylation of its downstream effector, eIF2α, progressively decrease during differentiation, and that the inhibition of PERK by GSK2606414 negatively impacts neutrophil differentiation. Interestingly, in BTHS cultured neutrophils, we observed elevated levels of p-eIF2α in mature cells, further confirming dysregulation of UPR in these cells. Neutropenia is one of the symptoms of Wolcott-Rallison syndrome caused by mutations in PERK, encoded by the *EIF2AK3* gene (Lundgren et al, 2019), and heightened ER stress was reported in neutropenias caused by mutations in *ELANE* and glucose-6-phosphate subunit α gene (*G6PC3*) (Boztug et al, 2009; Nanua et al, 2011; Sapra et al, 2020). These findings collectively highlight the importance of tightly regulated UPR in neutrophil development. Interestingly, supplementation with a PERK activator also led to dysregulation of neutrophil development, indicating that potential therapeutic strategies modulating this pathway may be complicated by the need for precise temporal control of interventions. Future studies will focus on defining the optimal developmental window for modulating PERK activity in developing neutrophils.

In summary, we confirmed a partial delay in neutrophil maturation in BTHS patients, resulting in generation of fewer mature neutrophils from stem cells. Despite alterations in phenotype and function, BTHS neutrophils maintain their antimicrobial activity and display a hyperinflammatory phenotype. Potential causes of these changes could be alterations in the UPR and calcium signaling pathways, which are currently recognized as vital for neutrophil development and function. The reduction in differentiation in the isolated neutrophil culture system does not fully recapitulate the severe neutropenia observed in some patients. BTHS neutropenia has irregular patterns, even in individual patients, and we speculate that further developmental disruptions could be triggered under non-homeostatic conditions such as metabolic perturbations or other forms of cellular stress, as well as enhanced by modulator mutations. These conundrums imply the presence of underlying imbalances alongside functional impaired neutrophil precursors, necessitating further research to understand the nuanced mechanisms at play.

# Methods

**Reagents and tools table**

| Reagent/resource | Reference or source | Identifier or catalog number |
| --- | --- | --- |
| **Experimental models** | | |
| **Recombinant DNA** | | |
| pLV[shRNA]-EGFP-U6>hTAFAZZIN[shRNA#1] | VectorBuilder | VB230419-1321bcj |
| pLV[shRNA]-EGFP-U6>hTAFAZZIN[shRNA#2] | VectorBuilder | VB230419-1322rmk |
| pLV[shRNA]-EGFP-U6>hTAFAZZIN[shRNA#3] | VectorBuilder | VB230419-1323pmk |
| pLV[shRNA]-EGFP-U6>Scramble_shRNA#1 | VectorBuilder | VB010000-0001mty |
| **Antibodies** | | |
| APC anti-human CD66b | BioLegend | 305118 |
| Alexa Fluor® 700 anti-human CD15 (SSEA-1) | BioLegend | 301920 |
| FITC anti-human CD16 | BioLegend | 302006 |
| PE anti-human CD10 | BioLegend | 312204 |
| Brilliant Violet 421™ anti-human CD63 | BioLegend | 353030 |
| PE/Cyanine7 anti-human CD62L | BioLegend | 304822 |
| PerCP/Cyanine5.5 anti-human CD101 | BioLegend | 331016 |
| Rabbit anti-tafazzin | Abcam | AB307148-1001 |
| Rabbit anti-GAPDH | Cell Signaling Technology | 2118S |
| Rabbit anti-phospho-eIF2α (D968) | Cell Signaling Technology | 3398T |
| Rabbit anti-PERK | Cell Signaling Technology | 3192T |
| Rabbit anti-IRE1α | Cell Signaling Technology | 3294T |
| Rabbit anti-BiP | Cell Signaling Technology | 3177T |
| Rabbit anti-HSP60 | Cell Signaling Technology | 12165S |
| Mouse anti-actin | Sigma-Aldrich | A4700-100UL |

| Reagent/resource | Reference or source | Identifier or catalog number |
|---|---|---|
| **Oligonucleotides and other sequence-based reagents** | | |
| hTAFAZZIN shRNA#1: TGCTTCCTCAGTTACACAAAGCTCGAGCTTTGTGTAACTGAGGAAGCA | VectorBuilder | VB230419-1321bcj |
| hTAFAZZIN shRNA#2: CTGTGGCATGTCGGAATGAATCTCGAGATTCATTCCGACATGCCACAG | VectorBuilder | VB230419-1322rmk |
| hTAFAZZIN shRNA#3: CGGACTTCATTCAAGAGGAATCTCGAGATTCCTCTTGAATGAAGTCCG | VectorBuilder | VB230419-1323pmk |
| scramble_shRNA: CCTAAGGTTAAGTCGCCCTCGCTCGAGCGAGGGCGACTTAACCTTAGG | VectorBuilder | VB010000-0001mty |
| hTAFAZZIN gRNA#1: TAZ + 154413511: UGCAGACAUCUGCUUCACCA | Synthego | |
| hTAFAZZIN gRNA#2: TAZ-154413490: GCAGAUGUCUGCAGCUGCAG | Synthego | |
| Non-targeting gRNA control#1: GCACUACCAGAGCUAACUCA | Synthego | |
| Non-targeting gRNA control#1: GUACGUCGGUAUAACUCCUC | Synthego | |
| **Chemicals, enzymes and other reagents** | | |
| CD34 MicroBead Kit, human | Miltenyi Biotec | 130-046-702 |
| EasySep™ Direct Human Neutrophil Isolation Kit | STEMCELL Technologies | 19666 |
| Histopaque®-1077 | Sigma-Aldrich | 10771 |
| Citrate-dextrose solution (ACD) | Sigma-Aldrich | 8013-89-6 |
| Human Flt3-ligand | Miltenyi Biotec | 130-096-479 |
| Human SCF | Miltenyi Biotec | 130-096-695 |
| Recombinant human IL-3 | Bio-Techne | 203-IL-010/CF |
| Human GM-SCF | Miltenyi Biotec | 130-093-864 |
| Human G-CSF | Miltenyi Biotec | 130-096-347 |
| TruCut™ Cas9 Protein v2 | Invitrogen | A36400 |
| P3 Primary Cell 4D-Nucleofector™ X Kit S | Lonza | V4XP-3032 |
| StemSpan™ SFEM | STEMCELL Technologies | 09650 |
| Human TruStain FcX™ | BioLegend | 422302 |
| FITC Annexin V | BioLegend | 640906 |
| Zombie Aqua™ fixable viability dye | BioLegend | 423102 |
| MitoTracker™ Red CMXRos | Invitrogen | M7512 |
| MitoTracker™ Green | Invitrogen | M46750 |
| tetramethylrhodamine, ethyl ester, perchlorate (TMRE) | Invitrogen | T669 |
| MitoSOX™ Red | Invitrogen | M36008 |
| X-Rhod-1, AM | Invitrogen | X14210 |
| Syto Green | Invitrogen | S7575 |
| N-Formyl-Met-Leu-Phe | Sigma-Aldrich | F3506-5MG |
| Phorbol 12-myristate 13-acetate | Sigma-Aldrich | P8139 |
| Calcium ionophore A23187 | Sigma-Aldrich | C7522 |
| Concavalin A | Sigma-Aldrich | C5275 |
| GSK 2606414 | Tocris | 5107 |
| human myeloperoxidase DuoSet ELISA | R&D Systems | DY3174 |
| human neutrophil elastase/ELA2 DuoSet ELISA | R&D Systems | DY9167-05 |
| human IL-1 beta/IL1F2 DuoSet ELISA | R&D Systems | DY201 |
| Seahorse XF Cell Mito Stress Test Kit | Agilent | 103015-100 |
| **Software** | | |
| FlowJo™ | | v10.81 |
| Fiji (ImageJ2) | | v1.51r |
| GraphPad Prism | | v10.2.0 |
| Adobe Illustrator 2022 | | v26.5 |
| **Other** | | |
| 4D-Nucleofector® X Unit | Lonza | AAF-1003X |
| LSRFortessa™ X-20 | BD Biosciences | |

| Reagent/resource | Reference or source | Identifier or catalog number |
|---|---|---|
| Seahorse XF96 Analyzer | Agilent | |
| FLUOstar Omega | BMG Labtech | |
| IncuCyte ZOOM™ | Essen BioScience | |
| EVOS® FL Auto Imaging System | ThermoFisher Scientific | |

**Table 1. Genetic information of patients.**

| Mutation | Effect | Exon/intron |
|---|---|---|
| c.9_10dupG | p.His4Alafs*130; frameshift | Exon 1 |
| c.51 G > A | p.Trp17*; nonsense | Exon 1 |
| c.82_84delGTG | p.Val28del; deletion | Exon 1 |
| c.118 A > G[a] | p.Asn40Asp; missense | Exon 2 |
| c.149 T > C | p.Leu50Pro; missense | Exon 2 |
| c.182del | deletion/frameshift | Exon 2 |
| c.207 C > G | p.His69Gln; missense | Exon 2 |
| c.216 C > A | p.Cys72*; nonsense | Exon 2 |
| c.228_232del | deletion/frameshift | Exon 2 |
| c.239 G > A | p.Gly80Glu or RNA splicing; missense | Exon 3 |
| c.281 G > A[a] | p.Arg94His; missense | Exon 3 |
| c.346 G > C | p.Gly116Arg; missense | Exon 3 |
| c.547_548insdel | insertion-deletion | Exon 7 |
| c.553 A > G | new exonic splicing donor at codon Met185; p.Lys182Glnfs*4; frameshift | Exon 7 |
| c.581 G > A | p.Trp194*; nonsense | Exon 7 |
| c.583+5 G > A | GTAAGG > GTAAAG; RNA splicing | Intron 7 |
| c.589 G > A[b] | p.Gly197Arg; missense | Exon 8 |
| c.589 G > T | p.Gly197Trp; missense | Exon 8 |
| c.646+1del | deletion/frameshift | Exon 8 |
| c.809_812del | deletion/frameshift | Exon 11 |
| c.837_838delTC | p.Gln280Glyfs*30; frameshift | Exon 11 |

[a]Variant present in 2 patients.
[b]Variant present in 4 patients.
For one patient variant was unknown.
For more details on *TAFAZZIN* variants, please refer to Human TAFAZZIN Gene Variant Database at: https://www.barthsyndrome.org/research/tafazzindatabase.html.

## Human subjects and samples

The study was approved by NHS Research Ethics Committee (REC permit number 09/H0202/52). Written informed consent was received from all patients and healthy donors. The patient cohort consisted of 28 patients (27 male), while the controls consisted of 31 individuals (27 male). Samples from some patients and healthy donors were obtained multiple times. Patient genetic data (*TAFAZZIN* mutations) are listed in Table 1. BTHS patients were on a variable G-CSF treatment schedule, varying from daily to weekly: the most frequent regimes were three times weekly or alternate daily. Weekly doses ranged from 0.12 mcg/kg/day to

10.92 mcg/kg/day. Venous blood was collected in EDTA tubes (BD Biosciences). For shRNA and CRISPR/Cas9 experiments, HSPC were isolated from apheresis blood (NHSBT, Filton, Bristol, UK) with NHS REC approval (18/EE/0265).

## Mouse model and experiments

All mouse experiments were performed in accordance with UK Home Office regulations (Project License PP9886217), under the oversight of the Animal Welfare and Ethical Review Board (AWERB) of the University of Glasgow.

The *Tafazzin* knockout mice were generated using G4 embryonic stem cells (isolated from C57BL/6Ncr x 129S6/SvEvTac F1 mice). After germline transmission of the targeted allele mice were bred for at least 10 generation to FVB/NCrl mice.

## CD34[+] HSPC isolation and neutrophil culture

CD34[+] HSPC were isolated either from peripheral blood of consented healthy donors and BTHS patients or from apheresis blood and then cultured according to a modified protocol of Naveh et al, (Naveh et al, 2024). In brief, peripheral blood mononuclear cells (PMBCs) were isolated by density centrifugation using Histopaque®-1077 (Sigma-Aldrich) according to manufacturer's instructions, followed by red cell lysis (55 mM $NH_4Cl$, 0.137 mM EDTA, 1 mM $KHCO_3$, pH 7.5). CD34[+] cells were enriched with a human CD34 Microbead Kit (Miltenyi Biotec) according to manufacturer's protocol. For peripheral blood HSPC, the cells were cultured in StemSpan™ Hematopoietic Cell Media (STEMCELL™ Technologies) supplemented with 1% (v/v) penicillin-streptomycin (P/S, Sigma-Aldrich) for the first 4 days and from day 5 of the culture, cells were cultured in Iscove's Modified Dulbecco's Media (IMDM, Gibco™) supplemented with 1% P/S and 10% (v/v) heat-inactivated fetal bovine serum (FBS, Sigma-Aldrich). Cells isolated from apheresis blood were cultured in IMDM only. Cytokines were added at the indicated concentrations and days of culture: stem cell factor (SCF, 50 ng/mL; day 0–5 of culture), Flt-3 ligand (50 ng/mL; day 0–5 of culture), interleukin-3 (IL-3, 10 ng/mL; day 0–5 of culture), granulocyte-macrophage colony-stimulating factor (GM-CSF, 10 ng/mL; day 3–7 of culture), and granulocyte colony-stimulating factor (G-CSF, 10 ng/mL; day 7–14 of culture). All inhibitors or activators were added on day 3, 5, 7, 10 and 14 of the culture. All functional assays (unless stated otherwise) were completed between Day 17 and 19 of culture.

## CRISPR-mediated knockout of *TAFAZZIN*

CRISPR/Cas9 genome editing was performed on day 3 cultured neutrophils, by nucleofection of ribonuclear particles, as described

in (Naveh et al, 2024). Synthego's CRISPRevolution modified sgRNAs were designed using company's Knockout Guide Design. The sgRNAs sequences can be found in Reagents and tools table.

## shRNA-mediated knockdown of *TAFAZZIN*

shRNA knockdown was achieved using lentiviral transduction of day 3 cultured HSPC. HEK293T cells were transfected with psPAX2 and pMD2.G packaging vectors, along with U6-based shRNA lentiviral vectors, with enhanced green fluorescent protein (eGFP) as a marker (VectorBuilder Inc.). Virus was concentrated using Lenti-X™ Concentrator and added to the cultured neutrophils. The shRNA sequences used in this study can be found in Reagents and tools table.

## Flow cytometry analysis

The analysis was conducted as described in (Rice et al, 2023). Briefly, cells were stained with 0.1% Zombie™ live/dead stain (BioLegend) and incubated for 10 min in the dark at RT. Samples were washed, incubated in Fc receptor blocking solution Human TruStain FcX™ (BioLegend) diluted in MACS buffer (5 mM EDTA, 0.5% BSA in PBS) for 5 min, and then a mix of primary antibodies in MACS was added followed by 30 min incubation on ice. Samples then were washed twice with MACS buffer and fixed for 20 min with 2% formaldehyde in PBS. Cells were analyzed with BD LSRFortessa™ X-20 Cell Analyzer. OneComp eBeads™ (Invitrogen™) were used as a single-color compensation control. For Annexin V positivity, cells were stained with FITC-Annexin V (BioLegend). At least 10,000 events were recorded per sample, and then analyzed in FlowJo™ software (version 10). All data in the paper are presented as median fluorescence intensity (MFI). Detailed information on antibodies used for flow cytometry analysis can be found in Reagents and tools table.

## Western blot

Cells were lysed directly in 2x Bolt™ LDS sample buffer (Invitrogen™) supplemented with Halt™ Protease and Phosphatase Inhibitor Cocktail (Thermo Scientific™) and heated for 10 min at 70 °C. Samples were sonicated and then run on 4–12% Bolt™ Bis-Tris Plus Mini Protein gels (Invitrogen™), followed by transfer onto Immobilon®-FL PVDF membrane (MERCK), using a standard wet transfer protocol. The membrane was blocked with 5% milk, incubated with primary antibodies overnight, and on the following day with secondary antibodies for 1 h at RT. Blots were imaged using a LI-COR Odyssey® XF Imager. Detailed information on primary antibodies used for western blot analysis can be found in Reagents and tools table.

## PLB-985 cell culture and differentiation

PLB-985 were cultured in RPMI-1640 (Sigma-Aldrich) supplemented with P/S and 10% FBS. After lentiviral transduction, cells were differentiated by replacing the media with RPMI-1640 with P/S, 2.5% FBS, 0.5% DMF, 1× Nutridoma-CS (Roche) and culturing for 7 days.

## Neutrophil isolation, oxidative burst, and NET formation

Neutrophils were isolated using the EasySep™ Direct Human Neutrophil Isolation Kit (STEMCELL™ Technologies) as per the

manufacturer's instructions. ROS production was quantified with the luminol method, as previously described (Harbort et al, 2015). Chemiluminescence was recorded in 2-min intervals using a FLUOstar® Omega plate reader (BMG LABTECH). NET assays were performed as described in (Amulic et al, 2017). Cells were stimulated with 50 nM PMA, 300 nM fMLP, or 10 µM A23187. For NETosis assessment with EVOS® FL and Incucyte® ZOOM imaging systems, cells were seeded into clear-bottomed 96-well plates, the stimulant or vehicle was added at time point = 0, and cells were incubated for the relevant period of time. To measure NETosis, cells were stained with SYTOX (1 µM) dye and total green area per well or percent of SYTOX-positive cells were quantified. For immunocytochemical detection of NETs, neutrophils were incubated with A23187 or vehicle for 4 h, fixed with 4% formaldehyde for 10 min, and then stained with the 3D9 primary antibody (1:1000 dilution in 1% BSA/PBS-T) (Tilley et al, 2022), followed by staining with a secondary antibody conjugated to fluorochrome and Hoechst. At least 100 cells from different fields of view were counted per experiment, and cells positive for 3D9 were counted as NETs.

## MPO and NE ELISA

Plasma MPO and MPO/NE secreted by cultured neutrophils were quantified using Human Myeloperoxidase DuoSet ELISA (R&D Systems) and Human Neutrophil Elastaste/ELA2 DuoSet ELISA (R&D Systems), according to manufacturer's instructions.

## Degranulation

In experiments on circulating neutrophils, cells were stimulated with N-formylmethionine-leucyl-phenylalanine (fMLP, 300 nM) for 30 min or *Streptococcus pyogenes* MGAS10270 for 40 min. For experiments on HSPC-derived neutrophils, cells were stimulated with A23187 (2.5 µM) for 30 min. After stimulation, cells were stained and fixed according to the flow cytometry protocol detailed above.

## Bacterial killing assay

Neutrophils ($2.5 \times 10^6$) were resuspended in 500 µL HBSS and combined with $5 \times 10^5$ *Streptococcus pyogenes* MGAS10270 bacteria (Beres and Musser, 2007) in HBSS supplemented with 2 mM CaCl$_2$, 2 mM MgCl$_2$ and 10% pooled human serum (SEQUENS IVD). Samples were incubated at 37 °C with atmospheric CO$_2$ levels on a rotator. Aliquots (50 µL) of each sample were taken immediately and thereafter every hour for 4 h, and numbers of viable bacteria enumerated following plating onto Todd-Hewitt agar plates supplemented with 0.5% (w/v) yeast extract and incubation overnight at 37 °C, 5% CO$_2$. Killing efficiency was calculated as number of colonies with neutrophils/number of colonies with serum only.

## Fluorescent immunohistochemistry and mouse whole blood analysis

Lung and spleen sections of germline *Tafazzin*$^{-/-}$ mice, were stained for neutrophil abundance with a neutrophil-specific calgranulin antibody, as previously described (Cela et al, 2022).

Tissue sections were imaged with a Leica DMI6000 inverted epifluorescence microscope and analysis was performed with ImageJ Fiji. For whole blood analysis, mice were tail vein bled, and blood sample analysis was performed using the Procyte DX Hematology analyzer.

## Phagocytosis

Heat-killed *Streptococcus pyogenes* MGAS10270 ($1 \times 10^9$) were stained with pHrodo™ Phagocytosis Particle Labeling Kit for Flow Cytometry (Invitrogen™) following the manufacturer's protocol. Neutrophils ($2 \times 10^6$) were resuspended in 500 μL RPMI + Q and $2 \times 10^8$ pHrodo-stained *Streptococcus pyogenes* were added for an MOI = 100. Suspensions were incubated with rotation at 37 °C for 1 h, taking 100 μl aliquots at multiple time points. Cells were then washed once and analyzed using BD LSRFortessa™ X-20 Cell Analyzer.

## Seahorse metabolic flux analysis

Seahorse assay was performed according to the protocol in (Naveh et al, 2024) using Seahorse XFe96 microplates (Agilent) and Seahorse XF DMEM medium (Agilent).

## Mitochondrial labelling

To label mitochondria, cells were incubated with 25 nM tetramethylrhodamine, ethyl ester, perchlorate (TMRE, Invitrogen™), 5 μM MitoSox™ Red (Invitrogen™), or 5 nM MitoTracker™ green (Invitrogen™) in prewarmed HBSS. Cells were incubated with each dye for 20 min at 37 °C, washed with fresh media, resuspended in MACS buffer, and analyzed using BD LSRFortessa™ X-20 Cell Analyzer.

## Intracellular calcium

Levels of intracellular $Ca^{2+}$ were determined using the cell permeant calcium indicator X-Rhod-1, AM (Invitrogen™). Cells were incubated with 500 nM X-Rhod-1 in RPMI for 1 h at 37 °C, thoroughly washed with fresh media, resuspended in MACS buffer, and analyzed with BD LSRFortessa™ X-20 Cell Analyzer.

## Proteomics and bioinformatics analysis

Isolated peripheral blood neutrophils from patients and controls were lysed and labeled with tandem mass spectrometry reagents (Thermo Fisher), as previously described (Rice et al, 2023). Peptides were identified by nano LC-MS/MS with a Orbitrap™ Fusion Tribrid™ Mass Spectrometer (Thermo Scientific™). Raw files were analyzed using Proteome Discoverer™ software v. 2 and cross-referenced against the human UniProt database (human). The protein groups were reassessed by an in-house script which initially selects a master protein by ID and quantitation metrics, then by annotation quality of Uniprot accessions. Data were $\log_2$ transformed and tested for statistical significance using Welch's $t$ test. IPA analysis was performed with a filter of $P < 0.05$ to identify biological trends in the proteins that were statistically significant between conditions. PCAs were calculated using the PCA function in the FactoMineR package and plotted using either ggplot (2D) or Plotly (3D).

## Statistical analysis

The analysis was performed using GraphPad Prism 10 software. Data are presented as mean ± SD. Unless stated otherwise, "n" refers to the number of biological repeats. Statistical analysis was conducted following a Shapiro–Wilk normality test. For comparisons between two samples, an unpaired two-tailed $t$ test was applied, whereas one-way or two-way ANOVA followed by post hoc test was used for comparisons involving multiple samples.

# Data availability

The mass spectrometry proteomics data have been deposited to the ProteomeXchange Consortium via the PRIDE (Perez-Riverol et al, 2022) partner repository with the dataset identifier PXD052714.

The source data of this paper are collected in the following database record: biostudies:S-SCDT-10_1038-S44319-025-00393-w.

# Peer review information

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

## Acknowledgements

We thank all the patients and blood donors for participating in our study, as well as nurses and clinical staff at the NHS Barth Syndrome Service and Bristol Children's Hospital. We acknowledge the assistance and support from Gillian Alexander, Maria Pelidis, Effie Chronopoulou, Rachel Schwartz, Olivia Gordon, Sally Turner, and Germaine Pierre. We also thank Jane Brittan for technical assistance with the bacterial experiments. This work was funded by Barth Syndrome Foundation (Idea Grants to BA and CS), by Bristol & Weston Hospitals Charity (through funds provided by the COGENT Trust), as well as MRC grant MR/R02149X/1 to BA. KF was funded by GW4-CAT Wellcome Trust Clinical PhD Fellowship. AT was funded by an NHS Blood and Transplant (NHSBT) R&D grant (WP15-05) and a National Institute for Health Research Blood and Transplant Research Unit (NIHR BTRU) in Red Blood Cell Products at the University of Bristol in partnership with NHSBT (IS-BTU-1214-10032). KR was funded by a Wellcome Trust Dynamic Cell PhD studentship. DS was funded by CRUK Scotland Institute core funding (A31287). The views expressed are those of the authors and not necessarily of the NHS, the NIHR or the Department of Health. The schematic summary figure was created in BioRender (https://BioRender.com/f19t736).

## Author contributions

**Przemysław Zakrzewski:** Formal analysis; Investigation; Visualization; Writing—original draft; Writing—review and editing. **Christopher M Rice:** Formal analysis; Investigation. **Kathryn Fleming:** Resources; Formal analysis; Investigation. **Drinalda Cela:** Formal analysis; Investigation. **Sarah J Groves:** Formal analysis; Investigation. **Fernando M Ponce-Garcia:** Formal analysis; Investigation. **Willem Gibbs:** Formal analysis; Investigation. **Kiran Roberts:** Formal analysis; Investigation. **Tobias Pike:** Formal analysis; Investigation. **Douglas Strathdee:** Resources. **Eve Anderson:** Formal analysis; Investigation. **Angela H Nobbs:** Supervision. **Ashley M Toye:** Conceptualization; Supervision; Funding acquisition. **Colin Steward:** Conceptualization; Supervision; Funding acquisition. **Borko Amulic:** Conceptualization; Formal analysis; Supervision; Funding acquisition; Writing—original draft; Writing—review and editing.

Source data underlying figure panels in this paper may have individual authorship assigned. Where available, figure panel/source data authorship is listed in the following database record: biostudies:S-SCDT-10_1038-S44319-025-00393-w.

## Disclosure and competing interests statement

The authors declare no competing interests.

# Expanded View Figures

**Figure EV1.  Additional analyses of BTHS neutrophils cultured ex vivo.**

(**A**) Representative image of BTHS patient bone marrow aspirate stained with Wright-Giemsa. (**B**) Quantification of average percentage of apoptotic (Annexin V + PI-) HSPC-derived neutrophils at the end of differentiation, $n = 2$ (HC), 3 (BTHS). (**C**) Representative cytospins of HSPC-derived neutrophils (day 17). (**D**) Simplified differential count of cytospins of HSPC-derived neutrophils (day 17), $n = 7$ (HC), $n = 7$ (BTHS). (**E**) Gating strategy for HSPC-derived neutrophils (day 17). (**F**) Average number of CD66b$^+$CD15$^+$ neutrophils in HSPC-derived cells at the of differentiation (day 17) per 1000 of HSPC cells at the start (day 0), $n = 8$ (HC), 9 (BTHS). (**G**). Average percentage of CD66b$^+$CD15$^+$CD11$^+$ cells in live population of HSPC-derived neutrophils at day 17, $n = 8$ (HC), 9 (BTHS); $P = 0.0013$. (**H**) Average percentage of CD66b$^+$CD15$^+$CD16$^+$ cells in live population of HSPC-derived neutrophils at day 17, $n = 8$ (HC), 9 (BTHS); $P = 0.0029$. Data information: Data are presented as mean ± SD. ns—not significant, **$P \leq 0.01$, assessed by unpaired $t$ test (**B**, **F–H**) and two-way ANOVA (**D**). Scale bar: 40 μm (**A**), 20 μM (**B**).

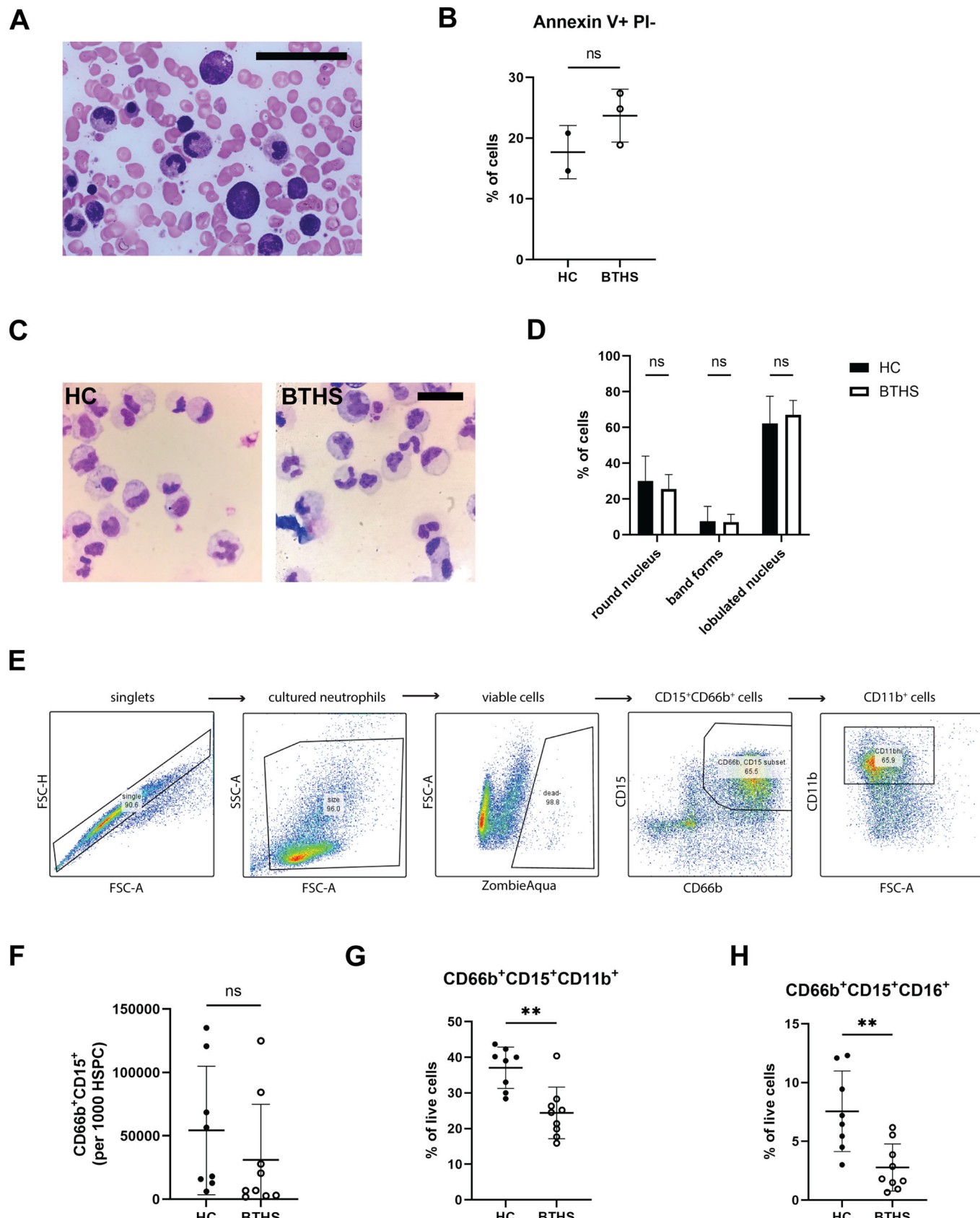

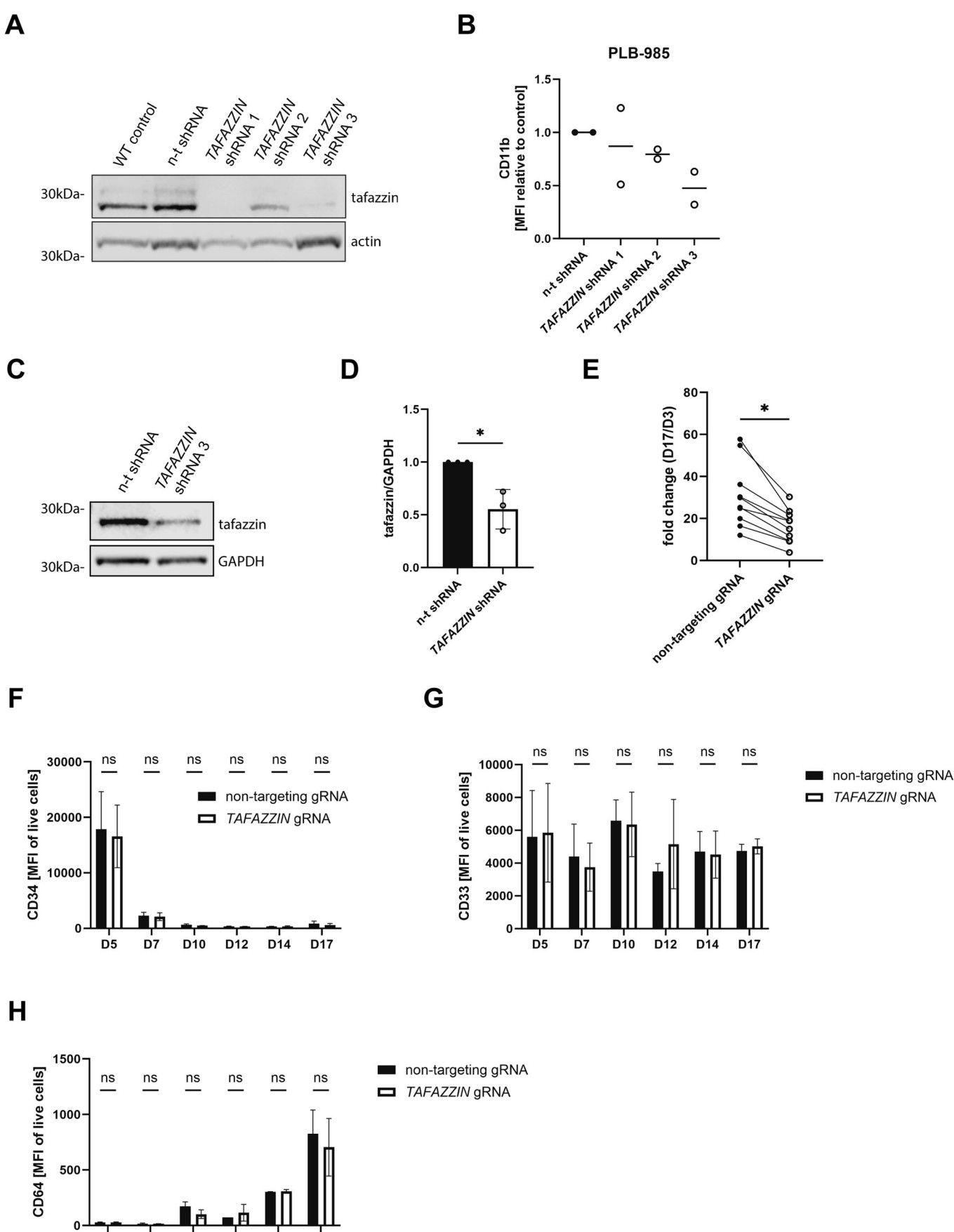

◀ **Figure EV2. Further characterization of tafazzin-deficient cells.**

(A) Representative western blot depicting level of tafazzin expression in PLB-985 cells transduced with lentivirus encoding non-targeting (n-t) or anti-*TAFAZZIN* shRNAs, $n = 2$ (experimental repeats). (B) Graph showing CD11b surface expression of PLB-985 cells transduced with lentivirus encoding non-targeting (n-t) or anti-*TAFAZZIN* shRNAs, $n = 2$ (experimental repeats). (C) Representative western blot depicting level of tafazzin expression level of GFP$^+$ HSPC-derived neutrophils transduced with lentivirus encoding non-targeting (n-t) or anti-*TAFAZZIN* shRNAs, $n = 3$. (D) Graph depicting level of tafazzin expression level of GFP$^+$ HSPC-derived neutrophils transduced with lentivirus encoding non-targeting (n-t) or anti-*TAFAZZIN* shRNAs, $n = 3$ (biological repeats); $P = 0.0146$. (E) Fold change of total cell count of CRISPR/Cas9-edited HSPC-derived neutrophils from day 3 to day 17 of differentiation matching *TAFAZZIN* knockout cells to their relative controls, $n = 10$ (biological repeats); $P = 0.0154$. (F) CD34 surface expression of HSPC-derived neutrophils (live population) during their differentiation in vitro, $n = 3$ (biological repeats). (G) CD33 surface expression of HSPC-derived neutrophils (live population) during their differentiation in vitro, $n = 3$ (biological repeats). (H) CD34 surface expression of HSPC-derived neutrophils (live population) during their differentiation in vitro, $n = 3$ (biological repeats). Data information: Data are presented as mean ± SD. ns—not significant, *$P \leq 0.05$, assessed by unpaired $t$ test (D, E) and two-way ANOVA (F–H).

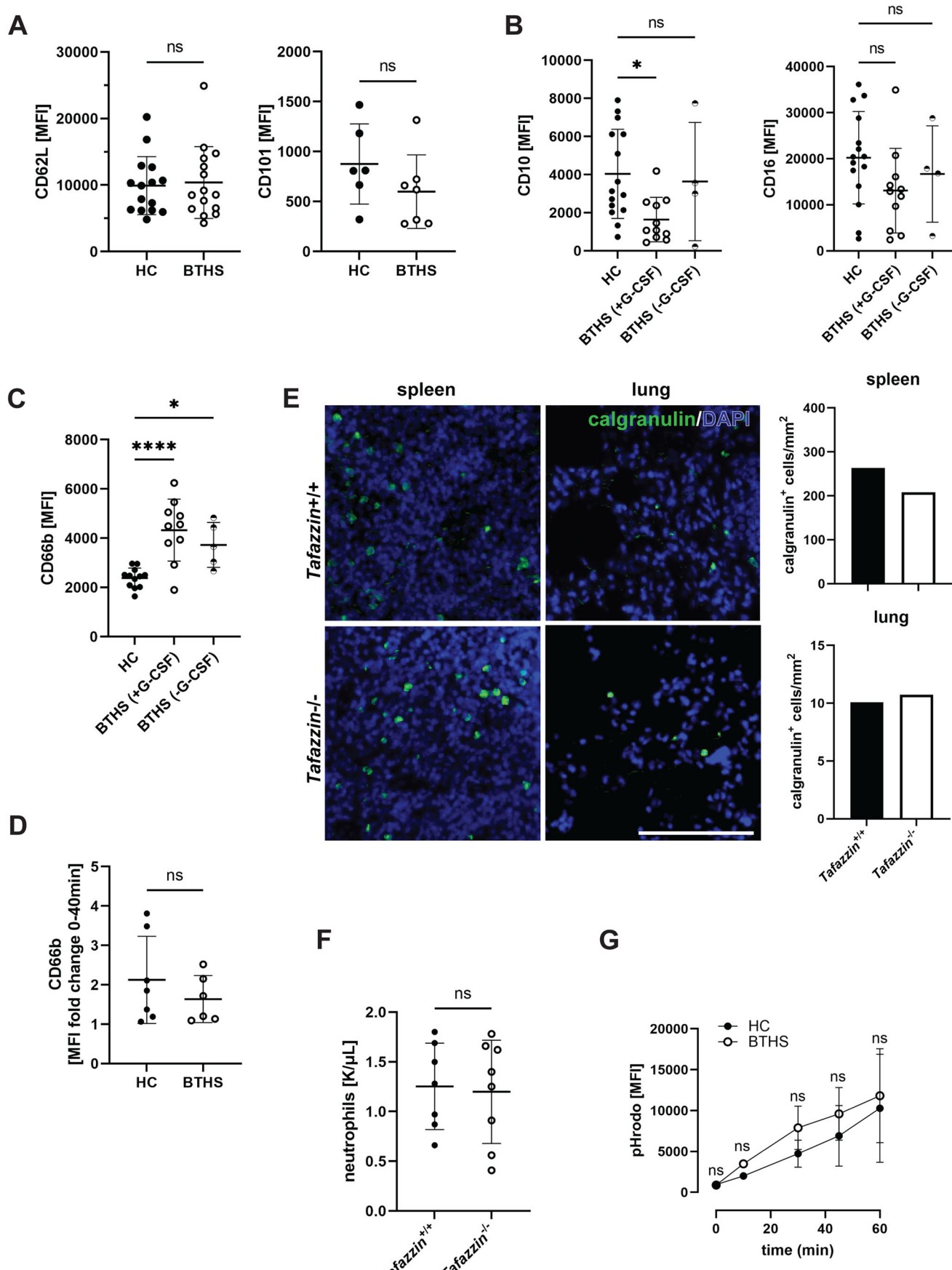

**Figure EV3. Additional analyses of peripheral blood neutrophils.**

(A) Surface expression of CD62L (left) and CD101 (right) in circulating neutrophils, $n = 15$ (HC, BTHS). (B) Surface expression of CD10 (left) and CD16 (right) in circulating neutrophils stratified according to G-CSF therapy; $n = 15$ (HC), 11 (BTHS + G-CSF), 4 (BTHS − G-CSF). (C) Surface expression of CD66b in neutrophils stratified according to G-CSF treatment, $n = 12$ (HC), 10 (BTHS + G-CSF), 5 (BTHS − G-CSF); p (HC vs. BTHS + G-CSF) = < 0.0001, p (HC vs. BTHS-G-CSF) = 0.0187. (D) Fold change of CD66b surface expression in isolated neutrophils stimulated with *Streptococcus pyogenes* for 40 min, MOI = 100, $n = 9$ (HC), 7 (BTHS). (E) Representative epifluorescence images showing anti-calgranulin (green) and DAPI (blue) staining of mouse lung and spleen (left) and quantification of average number of calgranulin-positive cells per mm² of section (right), $n = 2$ (mouse per genotype; the cells were counted from one transverse section through the middle part of the tissue); scale bar = 100 μm. (F) Absolute neutrophil counts in mouse whole blood, $n = 7$ (WT), 8 (KO). (G) Phagocytosis time-course, measured by pHrodo fluorescence in isolated neutrophils, $n = 3$ (HC, BTHS). Data information: Data are presented as mean ± SD. ns—not significant, *$P \leq 0.05$, ****$P \leq 0.0001$, assessed by unpaired $t$ test (A, D–G) and one-way ANOVA (B, C).

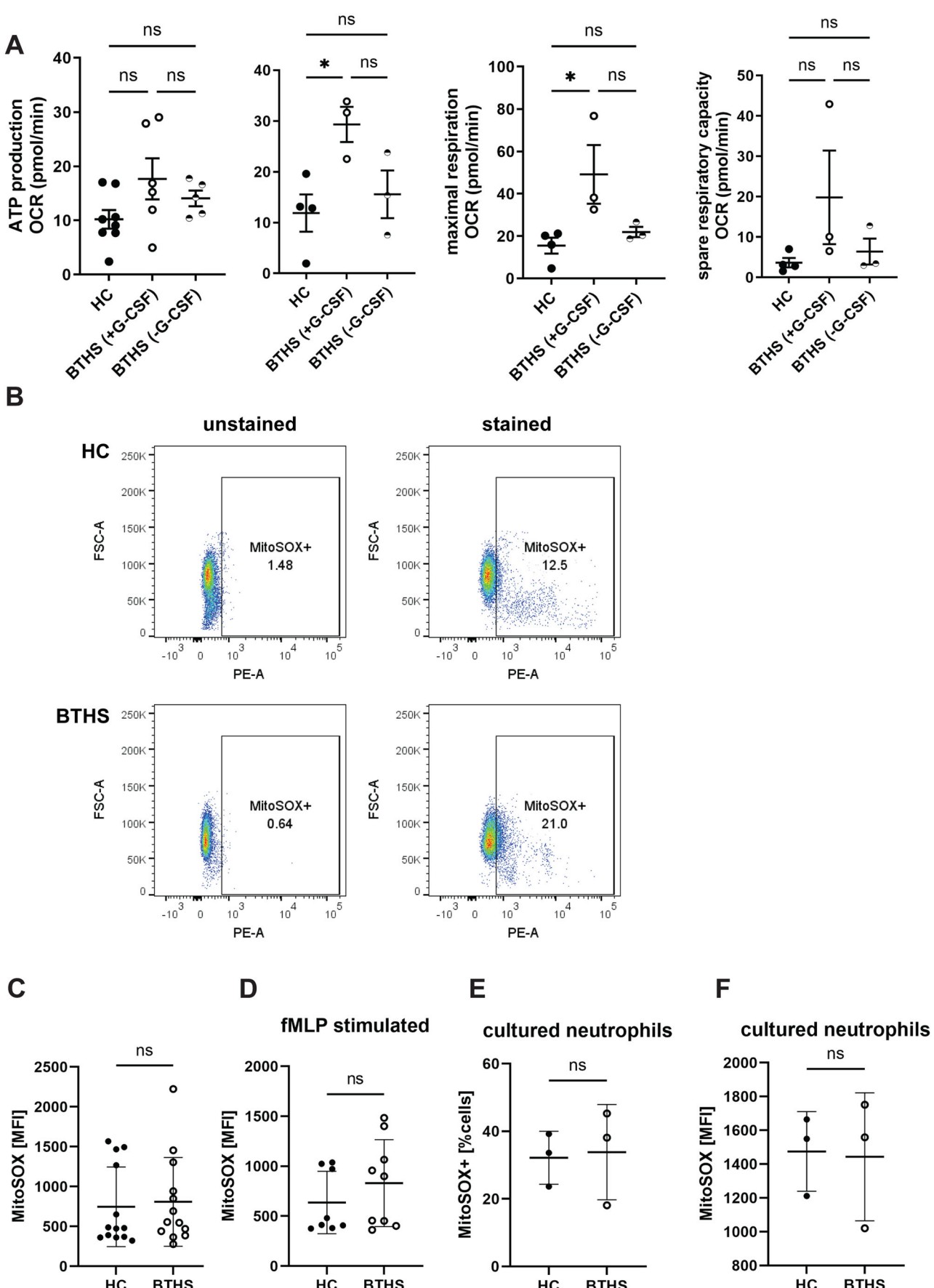

◀ **Figure EV4. Additional mitochondrial analyses in BTHS neutrophils.**

(A) Average rates of ATP production, basal respiration, maximal respiration, and spare respiratory capacity, measured by Seahorse metabolic flux analyzer and stratified according to G-CSF therapy; $n = 4$ (HC), 3 (BTHS + G-CSF), 3 (BTHS − G-CSF). (B) Gating strategy for MitoSOX+ circulating neutrophils. (C) Quantification of average MitoSOX median fluorescence of circulating neutrophils, $n = 13$ (HC, BTHS). (D) Quantification of average MitoSOX median fluorescence of circulating neutrophils stimulated with 300 nM fMLP, $n = 8$ (HC), 9 (BTHS). (E) Percentage of MitoSOX-positive HSPC-derived neutrophils at the end of differentiation (D17), $n = 3$ (HC, BTHS). (F) Percentage of MitoSOX-positive HSPC-derived neutrophils (D17), after stimulation with 300 nM fMLP, $n = 3$ (HC, BTHS). Data information: Data are presented as mean ± SD. ns—not significant, $^*P \leq 0.05$, assessed by one-way ANOVA (A) and unpaired $t$ test (C–F).

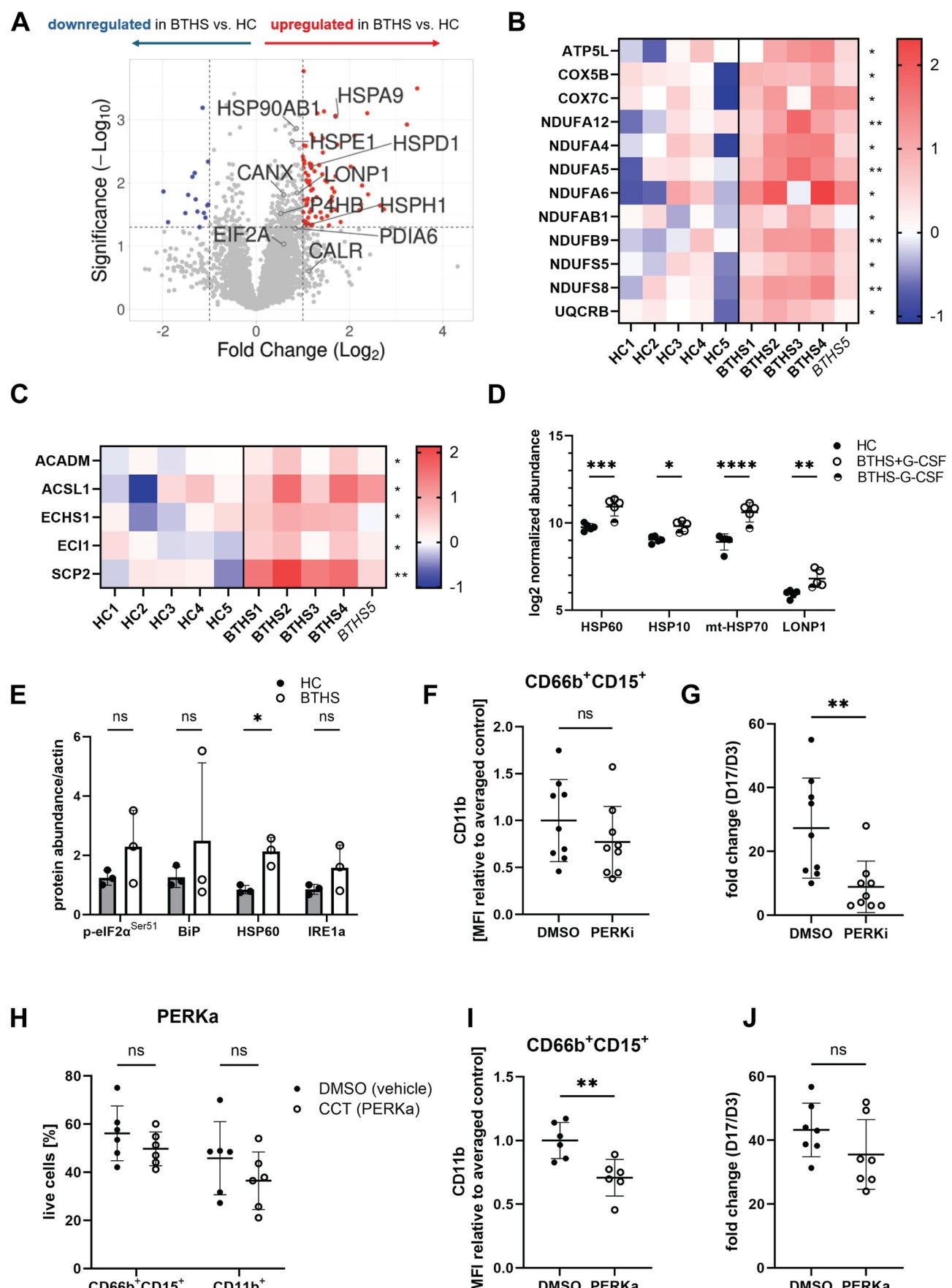

**Figure EV5. Elevated UPR signaling in BTHS neutrophils.**

(A) Volcano plot comparing BTHS and HC circulating neutrophil protein abundances, displayed as $-\log_{10}$ P-value and $\log_2$ fold change, $n = 5$ (HC), 5 (BTHS including $n = 4$ G-CSF-treated and $n = 1$ untreated, non-neutropenic patient). (B) Heat map depicting upregulated oxidative phosphorylation proteins identified with IPA, displayed as log2 fold change over averaged HC. (C) Heat map depicting upregulated fatty acid oxidation proteins identified with IPA, displayed as log2 fold change over averaged HC. (B, C) *BTHS5* - non-neutropenic BTHS patient. (D) $\log_2$ normalized abundance of mtUPR-related proteins identified by proteomics in circulating neutrophils, $n = 5$ (HC), 5 (BTHS including $n = 4$ G-CSF-treated and $n = 1$ untreated, non-neutropenic patient); p (HSP60) = 0.0002, p (HSP10) = 0.0192, P (mt-HSP60) = < 0.0001, P (LONP1) = 0.0064. (E) Quantification of UPR-related protein expression in HSPC-derived neutrophils at D17 of differentiation, $n = 3$ (HC, BTHS); P (HPS60) = 0.0106. (F). Surface expression of CD11b in PERK-inhibited (GSK2606414; 1 μM) or vehicle control (DMSO-treated) HSPC-derived neutrophils at D17 of differentiation, relative to averaged control (on a same day), $n = 9$ (DMSO, PERKi). (G) Fold change in total cell count during HSPC differentiation, from day 3 to day 17, after treatment with PERK inhibitor (GSK2606414; 1 μM) or vehicle control (DMSO), $n = 9$ (DMSO, PERKi); P = 0.0064. (H) Average percentage of HSPC-derived neutrophils (CD66b⁺CD15⁺) and mature neutrophils (CD66b⁺CD15⁺CD11b⁺) at D17, after treatment with PERK activator (CCT020312; 1 μM, added on day 3, 7, 10, and 14 of culture) or vehicle control (DMSO), $n = 6$ (DMSO, PERKa). (I) Surface expression of CD11b in PERK-activated (CCT020312; 1 μM) or vehicle control (DMSO-treated) HSPC-derived neutrophils at D17 of differentiation, relative to averaged control (on a same day), $n = 6$ (DMSO, PERKa); P = 0.0054. (J) Fold change in total cell count during HSPC differentiation, from day 3 to day 17, after treatment with PERK activator (CCT020312; 1 μM) or vehicle control (DMSO), $n = 7$ (DMSO, PERKa). Data information: Data are presented as mean ± SD. ns—not significant, *$P \le 0.05$, **$P \le 0.01$, ***$P \le 0.001$, ****$P \le 0.0001$, assessed by Welch's $t$ test after $\log_2$ transformation (B, C), unpaired $t$ test (D–G, I, J) and two-way ANOVA (H).

