## [Peer Review File · EMBO Reports]

Tafazzin regulates neutrophil maturation and inflammatory response

Przemyslaw Zakrzewski, Christopher Rice, Kathryn Fleming, Drinalda Cela, Sarah Groves, Fernando Ponce, Willem Gibbs, Kiran Roberts, Tobias Pike, Douglas Strathdee, Eve Anderson, Angela Nobbs, Ashley Toye, Colin Steward, and Borko Amulic

Corresponding author(s): Borko Amulic (borko.amulic@bristol.ac.uk)

Review Timeline:

Transfer Date:	6th Aug 24
Editorial Decision:	9th Aug 24
Revision Received:	19th Dec 24
Editorial Decision:	23rd Jan 25
Revision Received:	31st Jan 25
Accepted:	4th Feb 25

Editor: Achim Breiling

Transaction Report: This manuscript was transferred to EMBO reports following peer review at Review Commons.

**Review
COMMONS**

Revision Plan

Manuscript number: RC-2024-02552

Corresponding author(s): Amulic, Borko

1. General Statements [optional]
2. Description of the planned revisions

Please find below the point-by-point replies to the specific comments by the 3 reviewers (marked in bold).

Reviewer #1

Zakrzewski et al investigate the impact of tafazzin gene deficiency on neutrophils in patients with Barth Syndrome. Barth Syndrome is a rare X-linked genetic disease caused by mutations in the TFAZZIN gene and is characterized by pediatric-onset dilated cardiomyopathy, skeletal myopathy, and neutropenia. The researchers performed their studies on bone marrow aspirates, peripheral whole blood, and ex vivo differentiated neutrophils from hematopoietic stem cells isolated from a cohort of patients with Barth Syndrome (n=28). However, some concerns should be addressed before publication.

Major Concerns

1. When the authors analyze neutrophil degranulation, they find increased CD66b but no difference in CD63 exposure. Does that mean that neutrophils are degranulating secondary but not primary granules?

Patient neutrophils are spontaneously degranulating secondary granules (without stimulation) and show elevated rates of primary granule release upon stimulation. We will clarify this point in the text.

2. In lines 178-179 the authors find a "2-fold increase of CD66b upon bacterial stimulation". This is unclear from the graph. Does time zero in that graph represent the levels before stimulation? If so, it would seem that the levels are increased in the baseline, and not necessarily upon stimulation.

The reviewer is correct – there is increased levels of CD66B exposure at baseline, which indicates spontaneous degranulation in patients, compared to healthy control. This is an interesting and disease-relevant finding – we will clarify this point in the text.

3. What is the rate of degranulation over time upon stimulation? That would inform on the capacity of those neutrophils to mobilize granules.

Degranulation over time, in response to bacteria, is shown in Figure 2D. We agree that this is important data and we will highlight it in the revised manuscript.

4. Can the authors directly visualize and quantify the granules in their neutrophils to better characterise granularity? For instance, staining for MPO for primary granules.

This is a valuable suggestion. Instead of measuring markers of degranulation, we will quantify MPO by ELISA - the most direct and quantitative readout of primary degranulation.

5. In lines 197-198 the authors claim to find no difference in the number of neutrophils in the vasculature or parenchyma, indicating normal levels of margination (and Fig. S2D). This, unfortunately, cannot be concluded from the data, as the authors would need to control for the absolute number of neutrophils in circulation. If one of the genotypes has 10x more neutrophils in circulation and they find the same number of marginated neutrophils, then that would suggest 10x reduced margination capacity. Hence, controlling for circulating numbers is important in these experiments. The authors note that these mice have equivalent neutrophil counts, but then in Fig. S2E they show neutrophil ratios, not counts. Additionally, margination would refer to intravascular immobilization of neutrophils, but the stainings don't show the vasculature. How was the "number of neutrophils in the vasculature" (line 198) calculated?

We agree – we included absolute counts, rather than ratios in the revised manuscript (please see section 2).

In Fig S2E, we are quantifying the abundance of neutrophils in tissue (as opposed to circulation). We will change the term from 'margination' (which, as the reviewer correctly

Revision Plan

points out refers to vascular association), to 'organ-resident neutrophils'. We will also provide more detail on quantification method.

6. As a general note, the authors tend not to report absolute cell counts, only cell frequencies, for peripheral whole blood analyses. Changes in cell frequencies can be unintentionally misleading since a change can be due to a change in the numerator (i.e., cell population of interest) or the denominator (i.e., total cells). Reporting absolute cell counts is particularly important in this study since most of the Barth Syndrome patients are receiving G-CSF treatment, which on its own may alter the number and type of cells in circulation. If at all possible, the authors should report absolute cell counts.

We have provided absolute counts for mouse experiments (please see section 2), and will provide absolute numbers for ex vivo cultured BTHS patient neutrophils.

7. When the authors analyze the Tafazzin-KO mice they claim equivalent neutrophil counts (see previous notes about absolute numbers), and conclude that this suggests no defects in hematopoiesis. This, unfortunately, doesn't follow from the data shown. If the authors want to make this point, they should study the bone marrow hematopoietic stem and progenitor cell compartment in these mice.

Bone marrow analysis of tafazzin knockout mice has been reported by others and is discussed in the manuscript. We will remove this claim as suggested by the reviewer.

8. There is insufficient detail in the methods regarding the NET formation assays. How long were the neutrophils incubated with PMA? How do the authors differentiate NETs from necrotic neutrophils without any specific NET marker? Note that the authors show 15 to 20% necrotic neutrophils in their samples in Fig. S1B. In Fig 4E the authors show another NET quantification based on a DNA dye. The authors should consider showing representative images of their wells, and also to use a more specific way to show NET formation (i.e. looking for the colocalization of citrullinated histone 3, MPO (or Neutrophil Elastase) and DNA).

Revision Plan

We agree with the reviewer – we will provide more details on our dye-based quantification. Additionally, we will use a newly-developed NET-specific antibody (<https://elifesciences.org/articles/68283>), as a specific reagent to distinguish between NETosis and necrosis.

9. When the authors study mitochondrial abundance they find increased abundance in patients. Is that because these neutrophils are more immature? Then they find no changes in mitochondrial ATP production, basal respiration, maximal respiration or spare respiratory capacity. Therefore, they find more mitochondria but no differences in mitochondrial function. What does this mean?

We do in fact think that neutrophil immaturity accounts for elevated mitochondrial content. We will provide a detailed discussion on what we think this means.

10. Also, how is there an increased percentage of MitoSOX-positive cells with no difference in total MitoSOX MFI (lines 230-234)? What was the gating strategy for this experiment?

These two approaches measure related but distinct parameters. The paired analysis allows us to conclude that a larger proportion of mitochondria produce mitochondrial ROS. We will provide additional explanations and discussion to clarify this point, as well as the gating strategy.

Then, in lines 236-238 the authors write about mitochondria in G-CSF treated patients. Were these experiments performed only in samples from G-CSF-treated patients? That was unclear to this reviewer. As a general note, G-CSF treatment in the majority of Barth Syndrome patients is a significant confounder that is insufficiently addressed in the manuscript. There are very few controls for the effect of G-CSF treatment itself. The authors should also describe the G-CSF treatment regimen in the methods section, including dosing, frequency, route of administration, etc. In general, it is unclear in the text when and what samples come from G-CSF-treated patients. For instance, in figure 4A (line 247) the authors want to determine if G-CSF causes the changes in calcium levels, but the reader didn't know these data came from G-CSGF-

Revision Plan

treated patients... Please, clarify throughout the text.

We agree that this point deserved more clarification. The majority of BTHS patients at our clinic routinely receive GCSF therapy for their neutropenia. To control for this, we include data on the minority of non-GCSF (non-neutropenic) patients, and we also have access to healthy donors treated with GCSF. We will include more details to make this easier to understand. Importantly, all the ex vivo differentiation experiments do not have GCSF as a confounding factor, and are an excellent way to make conclusions on cell intrinsic tafazzin effects. We will emphasize this in the revised manuscript.

11. In lines 289-290 the authors claim "confirmed overexpression of IRE1 α , BiP, HSP60 and enhanced phosphorylation of 290 eIF2 α ", but the S5E shows that only HSP60 is significant. Along these same lines, on lines 297-298 they claim reduced CD11b surface expression, but Fig. S5F shows no significant difference. If the changes are not significant, the authors perhaps want to state that in the text, to avoid misleading the readers.

We agree and we will re-write these sections to include statistical significance.

12. In lines 312-313, the authors claim that "these results implicate dysregulation of unfolded protein response in impairment of neutrophil development in BTHS." It is unclear to this reviewer how so. The authors show that UPR inhibitors decreased the total number of neutrophils and the rest of the parameters they analyze were not significant (line 309 "although neither were statistically significant"). There could be alternative explanations. For instance, UPR inhibitors may decrease neutrophil lifespan or survival. It seems that the conclusion is a bit far-fetched from the data.

We agree with the reviewer that this aspect of the manuscript could benefit from additional data. To address this comment, we will perform CRISPR/Cas9 deletion of PERK, a key executor of the UPR, in HSC-derived neutrophils. We will quantify neutrophil development (maturation markers and proliferation rates) and functional responses (degranulation and NETs). Results will be compared with tafazzin knockouts, which will allow us to better assess mechanistic links between tafazzin deficiency and UPR signaling.

13. In the discussion (line 318) the authors claim that the study "reveals that tafazzin has a cell-intrinsic role in regulating neutrophil differentiation". While the authors show interesting data, a deeper look at this would have been interesting. For instance, could they quantify neutrophil progenitors during their in vitro differentiation to see if there's a blockade and at what level? And if so, what is the fate of these progenitors? In the discussion, the authors could comment how the decrease in neutrophil differentiation (defined here as CD66b+CD15+) the authors observed can account for the pronounced neutropenia seen in Barth Syndrome patients.

We agree with this suggestion and will quantify progenitors in neutrophil differentiations from tafazzin deficient HSC, using a more extensive FACS panel. This analysis will include the additional markers CD33 and CD64, which can be used to quantify progenitors. We will also add more discussion on how reduced neutrophil differentiation can lead to neutropenia, as suggested by the reviewer.

Minor Concerns

14. In line 96 the authors claim an elevated number of promyelocytes and myelocytes in Figure 1A, but myelocyte numbers seem unchanged in that figure, please check the wording. In the same figure, it would be useful, if feasible, to confirm the increase (or decrease) of the different cell populations compared to the normal levels using statistics.

The reviewer was correct, therefore we changed the wording. As the sample data was limited, the statistics would not be feasible, that is why we opted to show normal range in Fig. 1A and 1B.

15. Figure S1B seems to suggest a difference in necrosis in the BTHS group, likely not significant due to the very low power of this experiment (n=2 for the HC group).

The reviewer is correct, this experiment is underpowered and inconclusive. We propose to remove it from the supplement, since the relevant analysis here is for apoptosis (Fig. S1B and 2F and G) (suggested from previous publications). The propidium iodide + events in this

Revision Plan

figure could be either NETs or necrosis, and a more appropriate analysis, which distinguishes between these two, will be performed as part of additional experiments for comment 8 of Reviewer 1.

16. In line 113, the authors suggest there is no morphological changes in BTHS neutrophils, but the image shown may suggest increased nuclear hypersegmentation. In any case, quantification of morphological parameters would be appreciated. For instance, the authors could count the number of nuclear lobes and compare that between samples as, otherwise, the claim of no morphological changes is difficult to verify.

We will carry out quantification of nuclear morphology as suggested.

17. The authors use CD11b as a maturation maker (i.e. in Figure 1G), but CD11b is also an activation marker. Hence, the claim in line 116 of "a 31% decrease in the relative expression of the maturity marker CD11b" is difficult to interpret, as it may suggest decreased activation of those neutrophils. The authors use different markers for mature neutrophils between their peripheral whole blood analysis (CD10 positivity) and ex vivo HSC-derived differentiated neutrophils (CD15 positivity). It is unclear why the authors do not use the classical CD10 marker to identify mature neutrophils in their ex vivo differentiated neutrophil population.

We agree with the reviewer that an additional maturity marker would be beneficial. However due to technical difficulties we're unable to incorporate CD10 into our current panel. Instead, we will include data on another accepted and frequently-used neutrophil maturity marker – CD16 – in tafazzin WT and KO neutrophils.

<https://www.frontiersin.org/journals/immunology/articles/10.3389/fimmu.2019.01912/full>

18. In Figure 1H, CD11b levels are normalized to n-t shRNA, as is the case for other panels. But here it seems that all controls are equal to 1, suggesting that the authors normalized without retaining the variability of the control. The authors should normalize both conditions by the average of the controls (that is, normalize the controls to their own average) to retain the variability. Otherwise, the statistics are misleading because the variability in the controls is lost. The same is true for Figure 1K.

Revision Plan

We will make the suggested change for CRISPR experiments where we can normalise the data to averaged controls on a day of experiment. For the n-t shRNA experiment, we performed experiment 3 times using single n-t shRNA, therefore, we cannot normalise to average, hence all controls equal to 1.

19. In Figure 2C the individual data points are missing, and in Figure 3E the y-axis starts in negative numbers.

We will make the suggested changes.

20. In Figure S5A, it would be helpful for the readers to indicate at the top of the volcano plot which direction is what condition.

We will make the suggested changes

21. In line 301 the authors claim that IRE1a inhibitor did not affect development with "data not shown". The authors should either show the data or remove the claim.

We will remove this claim from the manuscript.

22. In line 309, Fig. 5H refers to Fig S5H, I believe, and in line 311, Fig 5I refers to S5I.

We will make the required change.

Reviewer #2

The study investigates the role of TFAZZIN in neutrophil maturation and inflammatory response in Barth syndrome (BTHS). Using a combination of bone marrow analysis, CRISPR/Cas9 genome editing, and functional characterization of neutrophils, the authors demonstrate a partial cell-intrinsic differentiation defect and dysregulated inflammatory response in BTHS. They suggest perturbations in the unfolded protein response (UPR) signaling pathway as a potential cause and therapeutic target.

Revision Plan

Major Comments:

The claims and conclusions are generally supported by the data presented. However, some conclusions, particularly those related to the mechanisms underpinning the observed neutrophil defects, would benefit from additional experiments.

22. Add more healthy controls to enhance statistics.

We agree that some experiments could benefit from additional biological replicates. We have added repeats (control and tafazzin deficient) to Figures 4C, 4D and 4E.

23. It remains unclear how NETosis was quantitated and presented in the different figures. In Figure 2i, '% NET positive' is plotted and an unusually high percentage of HC and BTHS neutrophils respond to PMA (80-90% HC and BTHS PMN). Normally, PMA induces 40-50% of neutrophils of healthy donors to undergo NETosis. In Figure 4D and E, NETosis is quantitated as "total green area", assuming this is the sum of Sytox Green stained area. Sytox is a cell-impermeable DNA dye that will also stain dead cells, so this analysis method does not distinguish other forms of cell death from NETosis. Further, in Figure 4D PMA does not seem to significantly induce NET formation in controls, contrasting results from HC and BTHS patient neutrophils. This needs to be clarified and NETosis should be defined as Sytox stained areas larger than a set threshold (e.g. 300 μ m²), to include cells undergoing chromatin decondensation only.

We agree with the reviewer that the NETs assay can be better described, and we will add more details. We induce NETs with 100 nM PMA, conditions which are known to reach high levels of NET induction (80-90%)

Eg: please see following reference.

"Automatic quantification of *in vitro* NET formation"

<https://www.frontiersin.org/journals/immunology/articles/10.3389/fimmu.2012.00413/full>

Additionally, we will include immunofluorescence stains of NETs with a NET-specific reagent (the antibody 3D9), as requested by reviewer 1 (comment 8).

24. The mechanism linking TFAZZIN deficiency to UPR pathway activation and neutrophil dysfunction could be investigated further. This could involve more detailed molecular analyses, such as RNA sequencing or proteomics focused on the UPR pathway. Use of further inhibitors/ activators or genetic knockdown (given the results with PERKi and PERKa). Also, a more detailed discussion on potential therapeutic implications of targeting the UPR pathway in BTHS could be added.

We agree with the reviewer. We will perform CRISPR/Cas9 knockouts of the UPR kinase PERK, in hematopoietic stem cells, to confirm involvement in neutrophil development. In order to address the link between tafazzin deficiency and UPR activation, we will test the compound elamipretide in tafazzin knockout neutrophils. This compound has been proposed to reverse the mitochondrial defects in tafazzin deficiency, and should be a useful tool to test whether mitochondrial dysfunction leads to UPR dysregulation. Finally we will add more discussion text on the therapeutic potential of targeting UPR in Barth syndrome.

Reviewer #3

The authors report a biological study exploring the pathophysiology of neutropenia in Barth syndrome (BTHS) related to Tafazzin mutations.

It is the most larger study which has studied the impact of Tafazin on the granulopoiesis, as 28 patients and 31 controls were studied.

The impact of tafazin on granulopoiesis was studied by various the techniques.

The following techniques were used

- cytology of the bone marrow
- bone marrow culture spontaneously or with shRNA or after CRISPR/Cas9 deletion
- flow cytometry of circulating neutrophils
- functional study of neutrophils (degranulation / bacterial killing)
- mitochondrial function
- NETosis in neutrophils
- UPR signalling

Globally, this study offers a global view of the neutrophil function and maturation defect in BTHS.

Revision Plan

Many results are fairly presented and this work, quite unique, deserves to be accepted.

Minor points:

25. In the references, a mention to the French series which report the data of 22 patients is lacking (PMID: 23656970).

The historical references to the initial paper of Barth is surely important, but with regards to the neutropenia study, the reference of Kuypers, already cited, signed as last author by PG Barth, is much more pertinent. Noteworthy, many functional neutrophil studies are already, although with less patients, reported in this work.

We will include the reference.

Reviewer #3 (Significance (Required)):

We were a little disappointed however to see that the understanding of neutropenia in BTHS remains elusive. BTHS is obviously related to TFAZZIN variants and the gene is involved in the energetic metabolism of the cells. The role of this cardiolipin metabolism defect can be easily understood in the heart muscle defect which uses it as a source of energy. It is less understood in the neutrophil defect, as the neutrophil source of energy seems independent from cardiolipin energetic metabolism. This study offers a large body of evidence but did not address this question. However it is the best evidence possible in this field today.

3. Description of the revisions that have already been incorporated in the transferred manuscript

Reviewer 1, comment 5: In lines 197-198 the authors claim to find no difference in the number of neutrophils in the vasculature or parenchyma, indicating normal levels of margination (and Fig. S2D). This, unfortunately, cannot be concluded from the data, as the authors would need to control for the absolute number of neutrophils in circulation. If one of the genotypes has 10x more neutrophils in circulation and they find the same number of marginated neutrophils, then that would suggest 10x

Revision Plan

reduced margination capacity. Hence, controlling for circulating numbers is important in these experiments. The authors note that these mice have equivalent neutrophil counts, but then in Fig. S2E they show neutrophil ratios, not counts.

In Fig. S2E we replaced % of neutrophils in WBC fraction to absolute neutrophil counts in neutrophils (x1000)/uL of blood.

Reviewer 2 (comment 22): Add more healthy controls to enhance statistics.

We agree that some experiments could benefit from additional biological replicates. We have added repeats (control and tafazzin deficient) to Figures 4C, 4D and 4E.

4. Description of analyses that authors prefer not to carry out

Reviewer 1, comment 7: When the authors analyze the Tafazzin-KO mice they claim equivalent neutrophil counts (see previous notes about absolute numbers), and conclude that this suggests no defects in hematopoiesis. This, unfortunately, doesn't follow from the data shown. If the authors want to make this point, they should study the bone marrow hematopoietic stem and progenitor cell compartment in these mice.

Bone marrow analysis of tafazzin knockout mice has been reported by others and is discussed in the manuscript. We will remove this claim as suggested by the reviewer.

Dear Dr. Amulic,

Thank you for the transfer of your research manuscript from Review Commons to EMBO reports. I now went through your manuscript, the referee reports from Review Commons (attached again below) and your revision plan. The referees have several comments, concerns, and suggestions to improve the manuscript, indicating that a major revision of the manuscript is necessary to allow publication of the study.

Going through your revision plan, it seems that the referee points will be adequately addressed during revision. I thus invite you to revise your manuscript accordingly with the understanding that all concerns must be addressed in the revised manuscript and/or in a final detailed point-by-point response (as indicated in your revision plan). Acceptance of your manuscript will depend on a positive outcome of another round of review using the same set of referees.

It is EMBO reports policy to allow a single round of major revision only and acceptance of the manuscript will therefore depend on the completeness of your responses included in the next, final version of the manuscript.

- 1) a .docx formatted version of the final manuscript text (including legends for main figures, EV figures and tables), but without the figures included. Figure legends should be compiled at the end of the manuscript text.
- 2) individual production quality figure files as .eps, .tif, .jpg (one file per figure), of main figures (up to 8) and EV figures (up to 5). Please upload these as separate, individual files upon re-submission.

- 3) a final .docx formatted letter INCLUDING the reviewers' reports and your detailed point-by-point responses to their comments. As part of the EMBO Press transparent editorial process, the point-by-point response is part of the Review Process File (RPF), which will be published alongside your paper.

- 4) a complete author checklist, which you can download from our author guidelines (<https://www.embopress.org/page/journal/14693178/authorguide>). Please insert page numbers in the checklist to indicate where the requested information can be found in the manuscript. The completed author checklist will also be part of the RPF.

- 5) that primary datasets produced in this study (e.g. RNA-seq, ChIP-seq, structural and array data) are deposited in an

appropriate public database. If no primary datasets have been deposited, please also state this in a dedicated section (e.g. 'No primary datasets have been generated and deposited'), see below.

The accession numbers and database should be listed in a formal "Data Availability" section (placed after Materials & Methods) that follows the model below. This is now mandatory (like the COI statement). Please note that the Data Availability Section is restricted to new primary data that are part of this study. This section is mandatory. As indicated above, if no primary datasets have been deposited, please state this in this section

Data availability

8) Regarding data quantification and statistics, please make sure that the number "n" for how many independent experiments were performed, their nature (biological versus technical replicates), the bars and error bars (e.g. SEM, SD) and the test used to calculate p-values is indicated in the respective figure legends (also for potential EV figures and all those in the final Appendix). Please also check that all the p-values are explained in the legend, and that these fit to those shown in the figure. Please provide statistical testing where applicable. Please avoid the phrase 'independent experiment', but clearly state if these were biological or technical replicates. Please also indicate (e.g. with n.s.) if testing was performed, but the differences are not significant. In case n=2, please show the data as separate datapoints without error bars and statistics. See also: <http://www.embopress.org/page/journal/14693178/authorguide#statisticalanalysis>

9) Please also note our reference format:

10) We updated our journal's competing interests policy in January 2022 and request authors to consider both actual and perceived competing interests. Please review the policy <https://www.embopress.org/competing-interests> and update your competing interests if necessary. Please name this section 'Disclosure and Competing Interests Statement' and put it after the Acknowledgements section.

11) We now use CRediT to specify the contributions of each author in the journal submission system. CRediT replaces the author contribution section. Please use the free text box to provide more detailed descriptions and do NOT provide an author contributions section in the revised manuscript text file. See also guide to authors:

<https://www.embopress.org/page/journal/14693178/authorguide#authorshippinguidelines>

12) Please add scale bars of similar style and thickness to all microscopic images, using clearly visible black or white bars (depending on the background). Please place these in the lower right corner of the images themselves. Please do not write on or near the bars in the image but define the size in the respective figure legend.

13) Please make sure that all the funding information is also entered into the online submission system and that it is complete

and similar to the one in the acknowledgement section of the manuscript text file.

14) All Materials and Methods need to be described in the main text using our 'Structured Methods' format, which is required for all research articles. According to this format, the Materials and Methods section should include a Reagents and Tools Table (listing key reagents, experimental models, software and relevant equipment and including their sources and relevant identifiers), uploaded as separate file, followed by a Methods and Protocols section in which we encourage the authors to describe their methods using a step-by-step protocol format with bullet points, to facilitate the adoption of the methodologies across labs. More information on how to adhere to this format as well as downloadable templates (.doc or .xls) for the Reagents and Tools Table can be found in our author guidelines (section 'Structured Methods'):

15) Please add up to five keywords to the manuscript and order the manuscript sections like this, using these names: Title page - Abstract - Keywords - Introduction - Results - Discussion - Methods - Data availability section - Acknowledgements - Disclosure and Competing Interests Statement - References - Figure legends - Expanded View Figure legends

I look forward to seeing a revised version of your manuscript when it is ready. Please let me know if you have questions or comments regarding the revision.

Best,

Achim Breiling
Senior editor
EMBO reports

Referee #1:

Zakrzewski et al investigate the impact of tafazzin gene deficiency on neutrophils in patients with Barth Syndrome. Barth Syndrome is a rare X-linked genetic disease caused by mutations in the TFAZZIN gene and is characterized by pediatric-onset dilated cardiomyopathy, skeletal myopathy, and neutropenia. The researchers performed their studies on bone marrow aspirates, peripheral whole blood, and ex vivo differentiated neutrophils from hematopoietic stem cells isolated from a cohort of patients with Barth Syndrome (n=28). However, some concerns should be addressed before publication.

****Major Concerns****

- When the authors analyze neutrophil degranulation, they find increased CD66b but no difference in CD63 exposure. Does that mean that neutrophils are degranulating secondary but not primary granules? In lines 178-179 the authors find a "2-fold increase of CD66b upon bacterial stimulation". This is unclear from the graph. Does time zero in that graph represent the levels before stimulation? If so, it would seem that the levels are increased in the baseline, and not necessarily upon stimulation. What is the rate of degranulation over time upon stimulation? That would inform on the capacity of those neutrophils to mobilize granules. Can the authors directly visualize and quantify the granules in their neutrophils to better characterize granularity? For instance, staining for MPO for primary granules.

- In lines 197-198 the authors claim to find no difference in the number of neutrophils in the vasculature or parenchyma, indicating normal levels of margination (and Fig. S2D). This, unfortunately, cannot be concluded from the data, as the authors would need to control for the absolute number of neutrophils in circulation. If one of the genotypes has 10x more neutrophils in circulation and they find the same number of marginated neutrophils, then that would suggest 10x reduced margination capacity. Hence, controlling for circulating numbers is important in these experiments. The authors note that these mice have equivalent neutrophil counts, but then in Fig. S2E they show neutrophil ratios, not counts. Additionally, margination would refer to intravascular immobilization of neutrophils, but the stainings don't show the vasculature. How was the "number of neutrophils in the vasculature" (line 198) calculated?

- As a general note, the authors tend not to report absolute cell counts, only cell frequencies, for peripheral whole blood analyses. Changes in cell frequencies can be unintentionally misleading since a change can be due to a change in the numerator (i.e., cell population of interest) or the denominator (i.e., total cells). Reporting absolute cell counts is particularly important in this study since most of the Barth Syndrome patients are receiving G-CSF treatment, which on its own may alter the number and type of cells in circulation. If at all possible, the authors should report absolute cell counts.
- When the authors analyze the Tafazzin-KO mice they claim equivalent neutrophil counts (see previous notes about absolute numbers), and conclude that this suggests no defects in hematopoiesis. This, unfortunately, doesn't follow from the data shown. If the authors want to make this point, they should study the bone marrow hematopoietic stem and progenitor cell compartment in these mice.
- There is insufficient detail in the methods regarding the NET formation assays. How long were the neutrophils incubated with PMA? How do the authors differentiate NETs from necrotic neutrophils without any specific NET marker? Note that the authors show 15 to 20% necrotic neutrophils in their samples in Fig. S1B. In Fig 4E the authors show another NET quantification based on a DNA dye. The authors should consider showing representative images of their wells, and also to use a more specific way to show NET formation (i.e. looking for the colocalization of citrullinated histone 3, MPO (or Neutrophil Elastase) and DNA).
- When the authors study mitochondrial abundance they find increased abundance in patients. Is that because these neutrophils are more immature? Then they find no changes in mitochondrial ATP production, basal respiration, maximal respiration or spare respiratory capacity. Therefore, they find more mitochondria but no differences in mitochondrial function. What does this mean? Also, how is there an increased percentage of MitoSOX-positive cells with no difference in total MitoSOX MFI (lines 230-234)? What was the gating strategy for this experiment? Then, in lines 236-238 the authors write about mitochondria in G-CSF treated patients. Were these experiments performed only in samples from G-CSF-treated patients? That was unclear to this reviewer. As a general note, G-CSF treatment in the majority of Barth Syndrome patients is a significant confounder that is insufficiently addressed in the manuscript. There are very few controls for the effect of G-CSF treatment itself. The authors should also describe the G-CSF treatment regimen in the methods section, including dosing, frequency, route of administration, etc. In general, it is unclear in the text when and what samples come from G-CSF-treated patients. For instance, in figure 4A (line 247) the authors want to determine if G-CSF causes the changes in calcium levels, but the reader didn't know these data came from G-CSGF-treated patients... Please, clarify throughout the text.
- In lines 289-290 the authors claim "confirmed overexpression of IRE1 α , BiP, HSP60 and enhanced phosphorylation of 290 eIF2 α ", but the S5E shows that only HSP60 is significant. Along these same lines, on lines 297-298 they claim reduced CD11b surface expression, but Fig. S5F shows no significant difference. If the changes are not significant, the authors perhaps want to state that in the text, to avoid misleading the readers.
- In lines 312-313, the authors claim that "these results implicate dysregulation of unfolded protein response in impairment of neutrophil development in BTHS." It is unclear to this reviewer how so. The authors show that UPR inhibitors decreased the total number of neutrophils and the rest of the parameters they analyze were not significant (line 309 "although neither were statistically significant"). There could be alternative explanations. For instance, UPR inhibitors may decrease neutrophil lifespan or survival. It seems that the conclusion is a bit far-fetched from the data.
- In the discussion (line 318) the authors claim that the study "reveals that tafazzin has a cell-intrinsic role in regulating neutrophil differentiation". While the authors show interesting data, a deeper look at this would have been interesting. For instance, could they quantify neutrophil progenitors during their in vitro differentiation to see if there's a blockade and at what level? And if so, what is the fate of these progenitors? In the discussion, the authors could comment how the decrease in neutrophil differentiation (defined here as CD66b+CD15+) the authors observed can account for the pronounced neutropenia seen in Barth Syndrome patients.

****Minor Concerns****

- In line 96 the authors claim an elevated number of promyelocytes and myelocytes in Figure 1A, but myelocyte numbers seem unchanged in that figure, please check the wording. In the same figure, it would be useful, if feasible, to confirm the increase (or decrease) of the different cell populations compared to the normal levels using statistics.
- Figure S1B seems to suggest a difference in necrosis in the BTHS group, likely not significant due to the very low power of this experiment (n=2 for the HC group).
- In line 113, the authors suggest there is no morphological changes in BTHS neutrophils, but the image shown may suggest increased nuclear hypersegmentation. In any case, quantification of morphological parameters would be appreciated. For instance, the authors could count the number of nuclear lobes and compare that between samples as, otherwise, the claim of no morphological changes is difficult to verify.
- The authors use CD11b as a maturation maker (i.e. in Figure 1G), but CD11b is also an activation marker. Hence, the claim in line 116 of "a 31% decrease in the relative expression of the maturity marker CD11b" is difficult to interpret, as it may suggest decreased activation of those neutrophils. The authors use different markers for mature neutrophils between their peripheral whole blood analysis (CD10 positivity) and ex vivo HSC-derived differentiated neutrophils (CD15 positivity). It is unclear why the authors do not use the classical CD10 marker to identify mature neutrophils in their ex vivo differentiated neutrophil population.
- In Figure 1H, CD11b levels are normalized to n-t shRNA, as is the case for other panels. But here it seems that all controls are equal to 1, suggesting that the authors normalized without retaining the variability of the control. The authors should normalize both conditions by the average of the controls (that is, normalize the controls to their own average) to retain the variability. Otherwise, the statistics are misleading because the variability in the controls is lost. The same is true for Figure 1K.
- In Figure 2C the individual data points are missing, and in Figure 3E the y-axis starts in negative numbers.
- In Figure S5A, it would be helpful for the readers to indicate at the top of the volcano plot which direction is what condition.

- In line 301 the authors claim that IRE1a inhibitor did not affect development with "data not shown". The authors should either show the data or remove the claim.

- In line 309, Fig. 5H refers to Fig S5H, I believe, and in line 311, Fig 5I refers to S5I.

****Significance****

This reviewer applauds the researchers for tackling the etiology of neutropenia in a rare, understudied disease and appreciates the inherent challenge of translational research with low patient numbers.

Referee #2:

The study investigates the role of TFAZZIN in neutrophil maturation and inflammatory response in Barth syndrome (BTHS). Using a combination of bone marrow analysis, CRISPR/Cas9 genome editing, and functional characterization of neutrophils, the authors demonstrate a partial cell-intrinsic differentiation defect and dysregulated inflammatory response in BTHS. They suggest perturbations in the unfolded protein response (UPR) signaling pathway as a potential cause and therapeutic target.

****Major Comments:****

The claims and conclusions are generally supported by the data presented. However, some conclusions, particularly those related to the mechanisms underpinning the observed neutrophil defects, would benefit from additional experiments.

- Add more healthy controls to enhance statistics.

- It remains unclear how NETosis was quantitated and presented in the different figures. In Figure 2i, '% NET positive' is plotted and an unusually high percentage of HC and BTHS neutrophils respond to PMA (80-90% HC and BTHS PMN). Normally, PMA induces 40-50% of neutrophils of healthy donors to undergo NETosis. In Figure 4D and E, NETosis is quantitated as "total green area", assuming this is the sum of Sytox Green stained area. Sytox is a cell-impermeable DNA dye that will also stain dead cells, so this analysis method does not distinguish other forms of cell death from NETosis. Further, in Figure 4D PMA does not seem to significantly induce NET formation in controls, contrasting results from HC and BTHS patient neutrophils. This needs to be clarified and NETosis should be defined as Sytox stained areas larger than a set threshold (e.g. 300µm²), to include cells undergoing chromatin decondensation only.

- The mechanism linking TFAZZIN deficiency to UPR pathway activation and neutrophil dysfunction could be investigated further. This could involve more detailed molecular analyses, such as RNA sequencing or proteomics focused on the UPR pathway. Use of further inhibitors/ activators or genetic knockdown (given the results with PERKi and PERKa). Also, a more detailed discussion on potential therapeutic implications of targeting the UPR pathway in BTHS could be added.

****Significance: ****

The study offers significant insights into the molecular mechanisms of neutropenia in BTHS, highlighting the role of TFAZZIN and the UPR pathway. The strongest aspects include the comprehensive approach combining patient samples, genome editing, and functional analyses. However, the study could be improved by increasing sample size, providing more mechanistic details and exploring therapeutic interventions.

The research will interest a specialized audience, including researchers and clinicians working on mitochondrial diseases, neutrophil biology, and genetic disorders. It has implications for basic research and potential translational/clinical applications.

Referee #3:

The authors report a biological study exploring the pathophysiology of neutropenia in Barth syndrome (BTHS) related to Tafazzin mutations.

It is the most larger study which has studied the impact of Tafazin on the granulopoiesis, as 28 patients and 31 controls were studied.

The impact of tafazin on granulopoiesis was studied by various the techniques.

The following techniques were used

- cytology of the bone marrow
- bone marrow culture spontaneously or with shRNA or after CRISPR/Cas9 deletion
- flow cytometry of circulating neutrophils
- functional study of neutrophils (degranulation / bacterial killing)
- mitochondrial function
- NETosis in neutrophils
- UPR signalling

Globally, this study offers a global view of the neutrophil function and maturation defect in BTHS. Many results are fairly presented and this work, quite unique, deserves to be accepted.

****Minor points:****

In the references, a mention to the French series which report the data of 22 patients is lacking (PMID: 23656970). The historical references to the initial paper of Barth is surely important, but with regards to the neutropenia study, the reference of Kuypers, already cited, signed as last author by PG Barth, is much more pertinent. Noteworthy, many functional neutrophil studies are already, although with less patients, reported in this work.

****Significance: ****

We were a little disappointed however to see that the understanding of neutropenia in BTHS remains elusive. BTHS is obviously related to TFAZZIN variants and the gene is involved in the energetic metabolism of the cells. The role of this cardiolipin metabolism defect can be easily understood in the heart muscle defect which uses it as a source of energy. It is less understood in the neutrophil defect, as the neutrophil source of energy seems independent from cardiolipin energetic metabolism. This study offers a large body of evidence but did not address this question. However it is the best evidence possible in this field today.

Manuscript number: EMBOR-2024-60151V2

Corresponding author(s): Amulic, Borko

Please find below the point-by-point replies to the specific comments by the reviewers.

Reviewer #1

Major Concerns

1. When the authors analyze neutrophil degranulation, they find increased CD66b but no difference in CD63 exposure. Does that mean that neutrophils are degranulating secondary but not primary granules?

Patient neutrophils demonstrate spontaneous degranulation of secondary granules (without stimulation) and show elevated rates of primary granule release upon stimulation.

We have clarified the text to clearly distinguish between spontaneous and induced degranulation of primary and secondary granules in BTHS patient neutrophils (lines 175-197).

2. In lines 178-179 the authors find a "2-fold increase of CD66b upon bacterial stimulation". This is unclear from the graph. Does time zero in that graph represent the levels before stimulation? If so, it would seem that the levels are increased in the baseline, and not necessarily upon stimulation.

We thank the reviewer for the comment. We clarified the text to highlight that BTHS neutrophils exhibit increased spontaneous degranulation of secondary granules. Additionally, we have included Fig. EV3D to demonstrate that the observed difference in induced degranulation levels is due to the elevated baseline ($t = 0$) CD66b expression, as pointed out by the reviewer.

3. What is the rate of degranulation over time upon stimulation? That would inform on the capacity of those neutrophils to mobilize granules.

Degranulation over time, in response to bacteria, is shown in Figure 3C. Additionally, we calculated the degranulation rate in BTHS and HC primary neutrophils (Fig. EV3D), as

suggested by the reviewer, showing that there is no difference in degranulation rate but only a difference in degranulation initiation. We hypothesize that this demonstrates an elevated propensity for activation, but normal granule trafficking rates.

4. Can the authors directly visualize and quantify the granules in their neutrophils to better characterise granularity? For instance, staining for MPO for primary granules.

We thank the reviewer for this suggestion. Unfortunately, we were unable to characterize granularity of BTBS primary neutrophils due to lack of access to additional clinical samples. Instead, we quantified MPO and neutrophil elastase (NE) by ELISA in WT and tafazzin CRISPR mutant neutrophils, in response to calcium mobilization. This is shown in Fig. 5C-D

5. In lines 197-198 the authors claim to find no difference in the number of neutrophils in the vasculature or parenchyma, indicating normal levels of margination (and Fig. S2D). This, unfortunately, cannot be concluded from the data, as the authors would need to control for the absolute number of neutrophils in circulation. If one of the genotypes has 10x more neutrophils in circulation and they find the same number of marginated neutrophils, then that would suggest 10x reduced margination capacity. Hence, controlling for circulating numbers is important in these experiments. The authors note that these mice have equivalent neutrophil counts, but then in Fig. S2E they show neutrophil ratios, not counts. Additionally, margination would refer to intravascular immobilization of neutrophils, but the stainings don't show the vasculature. How was the "number of neutrophils in the vasculature" (line 198) calculated?

We thank the reviewer for this valuable comment. We have included absolute neutrophil counts to demonstrate that there is no difference between WT and KO mice (Fig. EV3F). We have also changed the term from 'margination' (which, as the reviewer correctly points out refers to vascular association), to 'organ-resident neutrophils', for sake of accuracy.

6. As a general note, the authors tend not to report absolute cell counts, only cell frequencies, for peripheral whole blood analyses. Changes in cell frequencies can be unintentionally misleading since a change can be due to a change in the numerator (i.e., cell population of interest) or the denominator (i.e., total cells). Reporting absolute cell counts is particularly important in this study since most of the Barth Syndrome patients are receiving G-CSF treatment, which on its own may alter the

number and type of cells in circulation. If at all possible, the authors should report absolute cell counts.

We thank the reviewer for pointing this out. We have added absolute neutrophils counts for the mouse experiment (Fig. EV3F) and for *ex vivo* cultured patient neutrophils (normalized to 1000 starting cells at Day 3 of differentiation; Fig. EV1F).

7. When the authors analyze the Tafazzin-KO mice they claim equivalent neutrophil counts (see previous notes about absolute numbers), and conclude that this suggests no defects in hematopoiesis. This, unfortunately, doesn't follow from the data shown. If the authors want to make this point, they should study the bone marrow hematopoietic stem and progenitor cell compartment in these mice.

We agree with the reviewer and have revised the wording to avoid making conclusions about the hematopoietic compartment. Bone marrow analysis of tafazzin knockout mice has been reported by others and is discussed in the manuscript: (lines 209-218).

8. There is insufficient detail in the methods regarding the NET formation assays. How long were the neutrophils incubated with PMA? How do the authors differentiate NETs from necrotic neutrophils without any specific NET marker? Note that the authors show 15 to 20% necrotic neutrophils in their samples in Fig. S1B. In Fig 4E the authors show another NET quantification based on a DNA dye. The authors should consider showing representative images of their wells, and also to use a more specific way to show NET formation (i.e. looking for the colocalization of citrullinated histone 3, MPO (or Neutrophil Elastase) and DNA).

We have added more details to the Methods section regarding the NET formation assays:

" For NETosis assessment with EVOS® FL and Incucyte® ZOOM imaging systems, the cells were seeded into clear-bottomed 96-well plate, the stimulant or vehicle was added at time point = 0, and then cells were incubated for the relevant period of time. To measure NETosis, cells were incubated with SYTOX (1uM) dye and total green area per well or percent of SYTOX-positive cells was quantified at different time points. For immunocytochemical detection of NETs, neutrophils were incubated with stimulus or vehicle for 4 hours, fixed with 4% formaldehyde for 10 min, and then stained with the 3D9 primary antibody (1:1000 dilution in 1% BSA/PBS-T) [32], followed by staining with a secondary antibody conjugated to fluorochrome and Hoechst. At least 100 cells from

different fields of view were counted per experiment, and cells positive for 3D9 were counted as NETs."

We agree with the reviewer about the lack of specificity of our SYTOX-based live cell imaging assay. PMA and A23187 are standard and well-characterized NET inducers, however we also included an immunocytochemical staining method using the NET-specific antibody 3D9 (<https://elifesciences.org/articles/68283>) to validate our result. The 3D9 antibody discriminates NETs from necrotic and apoptotic cell death, by detecting a histone modification that is specific for NETs. Although we were unable to repeat the NET assay on BTHS patient neutrophils, we have included data in CRISPR KO cells (Fig. 5G-H).

9. When the authors study mitochondrial abundance they find increased abundance in patients. Is that because these neutrophils are more immature? Then they find no changes in mitochondrial ATP production, basal respiration, maximal respiration or spare respiratory capacity. Therefore, they find more mitochondria but no differences in mitochondrial function. What does this mean?

We thank the reviewer for the comment. We do in fact think that neutrophil immaturity accounts for elevated mitochondrial content. This immaturity phenotype is consistent with our flow cytometry data, showing reduced CD10 and CD16 expression. These findings align with the known effects of G-CSF treatment, which mobilizes immature neutrophils, although there is no direct evidence that immature neutrophils have more mitochondria. When possible, we stratified the data to distinguish the effects of G-CSF treatment from potential cell-intrinsic factors related to tafazzin deficiency. The stratified data further confirm that higher mitochondrial activity is limited to G-CSF-treated neutrophils, highlighting the role of this therapy in shaping the metabolic and developmental characteristics of these cells. We hope that corrected manuscript will help readers clearly distinguish between the effects attributable solely to G-CSF treatment and those caused by tafazzin deficiency.

The second part of the reviewer's question highlights an important finding: despite elevated mitochondrial content, mitochondrial respiration was unchanged. We hypothesize this suggests an underlying impairment that is 'hidden' at the cellular level because of increased total number of mitochondria. This is also consistent with elevated mitochondrial ROS in tafazzin deficient neutrophils: mitoROS production is often a hallmark of impaired electron transfer chain. We have emphasized this observation in the 250-264 section.

10. Also, how is there an increased percentage of MitoSOX-positive cells with no difference in total MitoSOX MFI (lines 230-234)? What was the gating strategy for this experiment?

We thank the reviewer for the comment, and we agree that this is an interesting contradiction. We speculate that this phenomenon arises from a higher percentage of BTHS neutrophils producing mtROS, although the overall range of intensities of MitoSOX staining per cell is similar or even lower to control. This might indicate that BTHS cells have a population of highly dysfunctional mitochondria (high mtROS producers) and suggests that mitochondria are not uniform and must be investigated at the population level, i.e. diversity of mitochondrial health must be considered per cell. We have revised the text to make this distinction clearer, please see lines 231-249 and 410-426

We have also included the gating strategy in Fig. EV4B.

Then, in lines 236-238 the authors write about mitochondria in G-CSF treated patients. Were these experiments performed only in samples from G-CSF-treated patients? That was unclear to this reviewer. As a general note, G-CSF treatment in the majority of Barth Syndrome patients is a significant confounder that is insufficiently addressed in the manuscript. There are very few controls for the effect of G-CSF treatment itself. The authors should also describe the G-CSF treatment regimen in the methods section, including dosing, frequency, route of administration, etc. In general, it is unclear in the text when and what samples come from G-CSF-treated patients. For instance, in figure 4A (line 247) the authors want to determine if G-CSF causes the changes in calcium levels, but the reader didn't know these data came from G-CSGF-treated patients... Please, clarify throughout the text.

The experiments referred to by the reviewer (lines 236-238 in original manuscript) were performed with a majority of samples from G-CSF-treated patients: MitoTracker experiment was performed with 7 GCSF-treated samples and 4 samples from non-treated (non-neutropenic) patients; TMRE experiment with 13 GCSF-treated patient samples and 4 non-treated (non-neutropenic) patients; and MitoSOX experiment on 12 GCSF-treated and 1 non-treated (non-neutropenic) sample).

We agree that this point deserves more clarification. The majority of BTHS patients at our clinic routinely receive G-CSF therapy for their neutropenia. Only the minority of non-neutropenic BTHS patients do not receive treatment and were also included in our study. We have revised the text to more clearly distinguish between the effects possibly attributable to G-CSF treatment and those caused by tafazzin deficiency.

As requested, we have also added details on G-CSF dosing in the methods section: "*BTHS patients were on a variable G-CSF treatment schedule, varying from daily to weekly: the most frequent regimes were three times weekly or alternate daily. Weekly doses ranged from 0.12 mcg/kg/day to 10.92 mcg/kg/day.*"

Importantly, our *ex vivo* HSPC differentiation experiments exclude G-CSF as a confounding factor, since concentration is fixed, and can be used to make robust conclusions on cell intrinsic tafazzin effects.

11. In lines 289-290 the authors claim "confirmed overexpression of IRE1 α , BiP, HSP60 and enhanced phosphorylation of 290 eIF2 α ", but the S5E shows that only HSP60 is significant. Along these same lines, on lines 297-298 they claim reduced CD11b surface expression, but Fig. S5F shows no significant difference. If the changes are not significant, the authors perhaps want to state that in the text, to avoid misleading the readers.

We appreciate this comment and have revised the text to clarify these points (lines: 348-351).

12. In lines 312-313, the authors claim that "these results implicate dysregulation of unfolded protein response in impairment of neutrophil development in BTHS." It is unclear to this reviewer how so. The authors show that UPR inhibitors decreased the total number of neutrophils and the rest of the parameters they analyze were not significant (line 309 "although neither were statistically significant"). There could be alternative explanations. For instance, UPR inhibitors may decrease neutrophil lifespan or survival. It seems that the conclusion is a bit far-fetched from the data.

We agree with the reviewer's comment and have revised the text to present a more cautious interpretation of our findings. Please see lines 348-351 and 427-449.

To further explore the involvement of UPR in neutrophil development, we attempted to use our CRISPR KO system to deplete maturing neutrophils of PERK. However, after five attempts, we were unable to confirm effective knockout of the gene. To further explore the potential role of PERK in neutrophil maturation and to refine our knockout approach, we performed a time-course analysis of PERK expression and downstream activity (phosphorylation of target protein eIF2alpha) using western blot (Fig. 6E). These data demonstrate that PERK expression and activity are both highest in early progenitors and are reduced with terminal neutrophil maturation. Future studies will focus on the role of PERK at these progenitor stages in WT and tafazzin $-/-$ cells.

13. In the discussion (line 318) the authors claim that the study "reveals that tafazzin has a cell-intrinsic role in regulating neutrophil differentiation". While the authors show interesting data, a deeper look at this would have been interesting. For instance, could they quantify neutrophil progenitors during their in vitro differentiation to see if there's a blockade and at what level? And if so, what is the fate of these progenitors? In the discussion, the authors could comment how the decrease in neutrophil differentiation (defined here as CD66b+CD15+) the authors observed can account for the pronounced neutropenia seen in Barth Syndrome patients.

We thank the reviewer for this thoughtful comment. We have performed additional characterization of progenitors in WT and tafazzin knockout cells, in order to investigate whether a blockade at an early stage of development is occurring. Using flow cytometry, we stained for additional markers of early progenitors (CD34, CD33, and CD64). These experiments did not reveal a clear blockade in neutrophil maturation (Fig. EV2F-H). We propose that, rather than an arrest at a particular stage, tafazzin depletion leads to a quantitative reduction in commitment to the neutrophil lineage. We have therefore revised the wording from "block" to "delay" in neutrophil differentiation and maturation, throughout the manuscript.

We also agree with the reviewer that the developmental delay that we observe in our *ex vivo* cultures does not fully explain the occurrence of severe neutropenia in some patients (although the neutropenia does range from mild to severe). We speculate that tafazzin deficiency leads to an inherent deficiency in myeloid commitment that is potentially exacerbated by metabolic or inflammatory stress, or genetic modulators, leading to severe neutropenia in some patients. Please see lines 455-463 in discussion.

Minor Concerns

14. In line 96 the authors claim an elevated number of promyelocytes and myelocytes in Figure 1A, but myelocyte numbers seem unchanged in that figure, please check the wording. In the same figure, it would be useful, if feasible, to confirm the increase (or decrease) of the different cell populations compared to the normal levels using statistics.

We thank the reviewer for pointing out this error. We changed the wording to reflect a change in promyelocytes only. As the sample data was limited and we did not have numerical data for healthy controls, we can only show normal range (reference clinical values) in Fig. 1A and 1B.

15. Figure S1B seems to suggest a difference in necrosis in the BTHS group, likely not significant due to the very low power of this experiment (n=2 for the HC group).

The reviewer is correct, this experiment is underpowered and inconclusive. We removed the necrosis analysis in original Fig. S1B and retained only the apoptosis analysis (Annexin V+ PI- staining), which is more relevant, and which is in line with the analysis showed in Fig. 3F and G. Another reason for excluding the PI+ quantification in this figure is that these could be either NETs or necrosis, and a more appropriate analysis, which distinguishes between these two, was performed in the new Fig. 5G-H).

16. In line 113, the authors suggest there is no morphological changes in BTHS neutrophils, but the image shown may suggest increased nuclear hypersegmentation. In any case, quantification of morphological parameters would be appreciated. For instance, the authors could count the number of nuclear lobes and compare that between samples as, otherwise, the claim of no morphological changes is difficult to verify.

We thank the reviewer for this valuable suggestion. We have performed differential counts of cytopins of neutrophils derived from BTHS patient and healthy HSPC, which is showed in Fig. EV1D. We categorized cells into three groups: cells with round, banded or lobulated nuclei – and we did not observe any difference between BTHS (n=7) and HC (n=7) samples (Fig. EV1D)

17. The authors use CD11b as a maturation maker (i.e. in Figure 1G), but CD11b is also an activation marker. Hence, the claim in line 116 of "a 31% decrease in the relative expression of the maturity marker CD11b" is difficult to interpret, as it may suggest decreased activation of those neutrophils. The authors use different markers for mature neutrophils between their peripheral whole blood analysis (CD10 positivity) and ex vivo HSC-derived differentiated neutrophils (CD15 positivity). It is unclear why the authors do not use the classical CD10 marker to identify mature neutrophils in their ex vivo differentiated neutrophil population.

We agree with the reviewer that an additional maturity marker would be beneficial. However, due to technical difficulties we're unable to incorporate CD10 into our flow cytometry panel for analyzing the *in vitro* differentiated cells. Instead, we have included data on another accepted and frequently-used neutrophil maturity marker – CD16 – in tafazzin-deficient and WT neutrophils.

<https://www.frontiersin.org/journals/immunology/articles/10.3389/fimmu.2019.01912/full>

For most experiments, we included CD16 MFI (Fig. 1H, 2B, 2F, whereas for experiments with patient-derived cultured neutrophils we have also included percentage of CD16-positive cells (Fig. EV1H).

18. In Figure 1H, CD11b levels are normalized to n-t shRNA, as is the case for other panels. But here it seems that all controls are equal to 1, suggesting that the authors normalized without retaining the variability of the control. The authors should normalize both conditions by the average of the controls (that is, normalize the controls to their own average) to retain the variability. Otherwise, the statistics are misleading because the variability in the controls is lost. The same is true for Figure 1K.

We appreciate the reviewer's comment pointing out the issue with how variability in the control group was represented. To address this concern and improve the robustness of the data, we have included additional repeats for our shRNA and CRISPR data and normalized

both the KO and control conditions to the averaged control MFIs when possible, thereby retaining variability in the control. We included these new datasets on the original graphs, ensuring that both the variability of the controls and the technical consistency of the data are accurately represented (Fig. 1H – new Fig. 2A, Fig. 1K – new Fig. 2E).

19. In Figure 2C the individual data points are missing, and in Figure 3E the y-axis starts in negative numbers.

We had tried to show individual data points in Fig. 2D (original Fig. 2C) but as the data points are numerous, they become merged and it makes the graph difficult to read (see below), which is why we opted to leave the bar chart. This is in line with journal formatting guidelines: *"If $n < 5$, please show single datapoints for diagrams"*

We thank the reviewer for noticing the negative values for spare respiratory capacity, which were incorrect and have been removed.

20. In Figure S5A, it would be helpful for the readers to indicate at the top of the volcano plot which direction is what condition.

We have made suggested changes to the plot to make it clearer (Fig. EV6).

21. In line 301 the authors claim that IRE1a inhibitor did not affect development with "data not shown". The authors should either show the data or remove the claim.

We removed this claim.

22. In line 309, Fig. 5H refers to Fig S5H, I believe, and in line 311, Fig 5I refers to S5I.

We corrected it and thank the reviewer for pointing it out.

Reviewer #2

The study investigates the role of TFAZZIN in neutrophil maturation and inflammatory response in Barth syndrome (BTHS). Using a combination of bone marrow analysis, CRISPR/Cas9 genome editing, and functional characterization of neutrophils, the authors demonstrate a partial cell-intrinsic differentiation defect and dysregulated inflammatory response in BTHS. They suggest perturbations in the unfolded protein response (UPR) signaling pathway as a potential cause and therapeutic target.

Major Comments:

The claims and conclusions are generally supported by the data presented. However, some conclusions, particularly those related to the mechanisms underpinning the observed neutrophil defects, would benefit from additional experiments.

22. Add more healthy controls to enhance statistics.

We have added more biological replicates to following CRISPR KO, shRNA, and PERK agonist experiments to enhance statistics: Fig. 2A-B, 2E-F, EV2E, 5E-F, EV6H-J.

23. It remains unclear how NETosis was quantitated and presented in the different figures. In Figure 2i, '% NET positive' is plotted and an unusually high percentage of HC and BTHS neutrophils respond to PMA (80-90% HC and BTHS PMN). Normally, PMA induces 40-50% of neutrophils of healthy donors to undergo NETosis. In Figure 4D and E, NETosis is quantitated as "total green area", assuming this is the sum of Sytox Green stained area. Sytox is a cell-impermeable DNA dye that will also stain

dead cells, so this analysis method does not distinguish other forms of cell death from NETosis. Further, in Figure 4D PMA does not seem to significantly induce NET formation in controls, contrasting results from HC and BTHS patient neutrophils. This needs to be clarified and NETosis should be defined as Sytox stained areas larger than a set threshold (e.g. 300µm²), to include cells undergoing chromatin decondensation only.

We agree with the reviewer that the NETs assay could have been better described and have added more details. We induce NETs with 100 nM PMA, conditions which are known to reach high levels of NET induction (80-90%); e.g.: please see following reference.

"Automatic quantification of *in vitro* NET formation"

<https://www.frontiersin.org/journals/immunology/articles/10.3389/fimmu.2012.00413/full>

Additionally, we have included quantification of NETs by immunofluorescence staining with a NET-specific reagent (the antibody 3D9), as explained in response to reviewer 1.

The NET-specific antibody, 3D9 (<https://elifesciences.org/articles/68283>), confirmed the phenotype obtained using SYTOX live cell imaging (Fig. 5G-H).

24. The mechanism linking TFAZZIN deficiency to UPR pathway activation and neutrophil dysfunction could be investigated further. This could involve more detailed molecular analyses, such as RNA sequencing or proteomics focused on the UPR pathway. Use of further inhibitors/ activators or genetic knockdown (given the results with PERKi and PERKa). Also, a more detailed discussion on potential therapeutic implications of targeting the UPR pathway in BTHS could be added.

To further explore the involvement of UPR in neutrophil development, we attempted to use our CRISPR KO system to deplete maturing neutrophils of PERK. However, after five attempts, we were unable to confirm effective knockout. To further explore the potential role of PERK in neutrophil maturation, we included a time-course analysis of PERK expression and activity using western blot (Fig. 6E). This demonstrated that PERK is abundantly expressed in myeloid progenitors, and that it gets activated (shown by phosphorylation of downstream target eIF2a), further strengthening the evidence for its involvement in regulation of neutrophil maturation.

We have also included additional text in the discussion, highlighting issues with targeting PERK and UPR in myelopoiesis (lines 444-449).

Reviewer #3

The authors report a biological study exploring the pathophysiology of neutropenia in Barth syndrome (BTHS) related to Tafazzin mutations.

It is the most larger study which has studied the impact of Tafazin on the granulopoiesis, as 28 patients and 31 controls were studied.

The impact of tafazin on granulopoiesis was studied by various the techniques.

The following techniques were used

- cytology of the bone marrow
- bone marrow culture spontaneously or with shRNA or after CRISPR/Cas9 deletion
- flow cytometry of circulating neutrophils
- functional study of neutrophils (degranulation / bacterial killing)
- mitochondrial function
- NETosis in neutrophils
- UPR signalling

Globally, this study offers a global view of the neutrophil function and maturation defect in BTHS.

Many results are fairly presented and this work, quite unique, deserves to be accepted.

Minor points:

25. In the references, a mention to the French series which report the data of 22 patients is lacking (PMID: 23656970).

The historical references to the initial paper of Barth is surely important, but with regards to the neutropenia study, the reference of Kuypers, already cited, signed as last author by PG Barth, is much more pertinent. Noteworthy, many functional neutrophil studies are already, although with less patients, reported in this work.

We have included the reference throughout the manuscript.

Reviewer #3 (Significance (Required)):

We were a little disappointed however to see that the understanding of neutropenia

in BTHS remains elusive. BTHS is obviously related to TFAZZIN variants and the gene is involved in the energetic metabolism of the cells. The role of this cardiolipin metabolism defect can be easily understood in the heart muscle defect which uses it as a source of energy. It is less understood in the neutrophil defect, as the neutrophil source of energy seems independent from cardiolipin energetic metabolism. This study offers a large body of evidence but did not address this question. However it is the best evidence possible in this field today.

Dear Dr. Amulic,

Thank you for the submission of your revised manuscript to our editorial offices. I have now received the report from one of the two referees that I asked to re-evaluate the study, you will find below. As you will see, the referee now basically supports the publication of the study in EMBO reports. Original referee #2 was completely unresponsive to my invitations to re-assess the study. However, going through your p-b-p-response, I consider the points of this referee as adequately addressed. Original referee #3 was already positive regarding the previous version of the manuscript, and I also consider his/her minor points as fully addressed.

I will thus proceed with the manuscript. However, I have these editorial requests I ask you to address in a final revised manuscript:

- Please add up to 5 keywords to the manuscript and order the sections like this, using these names:

Abstract - Keywords - Introduction - Results - Discussion - Methods - Data Availability Section - Acknowledgements - Disclosure and Competing Interests Statement - References - Figure legends - Expanded View Figure legends

- Please use the name 'Data Availability Section'. This section is restricted to information about externally deposited datasets. Thus, please remove the sentence 'The research materials supporting this publication can be accessed by contacting corresponding authors' from this section. Moreover, please remove now the referee access information from this section, add a direct link to the deposited dataset and make sure the data is public latest upon online publication of the paper.

- Please add all the relevant funding information to the Acknowledgements section.

- Please use the name 'Disclosure and Competing Interests Statement'.

- Please move the information regarding ethics approval and patient consent to the Methods section (Human subjects and samples).

- We now use CRediT to specify the contributions of each author in the journal submission system. CRediT replaces the author contribution section. Please use the free text box to provide more detailed descriptions and do NOT provide your final manuscript text file with an author contributions section. See also our guide to authors: <https://www.embopress.org/page/journal/14693178/authorguide#authorshipguidelines>

- Please use our reference format:

- Regarding data quantification and statistics, please check again that the number "n" for how many independent experiments were performed, their nature (biological versus technical replicates), the bars and error bars (e.g. SEM, SD) and the test used to calculate p-values are indicated in the respective figure legends (also for potential EV figures and all those in the final Appendix). Please also check that all the p-values are explained in the legend, and that these fit to those shown in the figure. Please provide statistical testing where applicable. Please also clearly state if these were biological or technical replicates. Please also indicate (e.g. with n.s.) if testing was performed, but the differences are not significant. In case n=2, please show the data as separate datapoints without error bars and statistics.

If n<5, please show single datapoints for diagrams. Could statistics be added to the diagrams shown in 1D, 3J, 6E and EV3G? Moreover:

- Please provide the exact p values are not provided in the legends of figures 1F, G; 2A, E, F; 3A, B, D, E, F; 4A, B, E; 5A, B, C, F, H; 6E, F; EV1 G, H; EV2 D, E; EV3 C; EV5 D, E, G, I;

- Please indicate the statistical test used for data analysis in the legends of figures 1E, F, G, H; 2A, B, E, F; 3A, B, D, E, F, G, H, I; 4A-E; 5A, B, C, D, E, F, H; 6A, B, C, F; EV1 B, D, F, G, H; EV1 B, D, F, G, H; 2F, G, H; EV3 A, C, D-F; EV4 A, C, D, E, F; EV5 F, H, J;

- Please add scale bars to the images shown in panels EV1A and C and define these in the legend.

- Please add to each legend (main and EV figures) a 'Data Information' section (or please use this name for the information already present) explaining the statistics used or providing information regarding replicates and scales. See:

- Please remove the instructions from the reagents and tools table and add callouts for the table to the methods section where applicable.

- Thank you for providing the requested source data. Please upload this as one folder per figure (with all files for one figure in one folder and ZIPed together).

In addition, I would need from you uploaded separately:

Referee #1:

The authors have answered most of my comments. This reviewer appreciates the effort the authors have put into revising the text and understands that obtaining further clinical material to conduct the requested experiments may be challenging. While this slightly reduces my enthusiasm for the manuscript, it remains a valuable effort to understand a severely understudied disease and, as such, may warrant publication in its current form at the Editor's discretion.

All editorial and formatting issues were resolved by the authors.

Borko Amulic
University of Bristol
School of Cellular and Molecular Medicine,
University Walk
Bristol
United Kingdom

Dear Dr. Amulic,

I am very pleased to accept your manuscript for publication in the next available issue of EMBO reports. Thank you for your contribution to our journal.

Yours sincerely,
